# Melting over the Northeast Antarctic Peninsula (1999-2009): evaluation of a high-resolution regional climate model

Rajashree T. Datta[1], Marco Tedesco[2], Cecile Agosta[3], Xavier Fettweis[3], Peter Kuipers Munneke[4], Michiel R. van den Broeke.[4]

[1]The Graduate Center, City University of New York, NY 10016, USA
[2]Lamont-Doherty Earth Observatory of Columbia University, Palisades, New York, New York 10964, USA
[3]Department of Geography, Université de Liège, Liège, Belgium
[4]Institute for Marine and Atmospheric Research, Utrecht University, Utrecht, The Netherlands

*Correspondence to*: Rajashree Tri Datta (Tri.Datta@gmail.com)

**Abstract.** Surface melting over the Antarctic Peninsula (AP) may impact the stability of ice shelves and therefore the rate at which grounded ice is discharged into the ocean. Energy and mass balance models are needed to understand how climatic change and atmospheric circulation variability drive current and future melting. In this study, we evaluate the regional climate model MAR over the AP at a 10 km spatial resolution between 1999 and 2009, a period when active microwave data from the QuikSCAT mission is available. This model has been validated extensively over Greenland, has is applied here to the AP at a high resolution and for a relatively long time period (full outputs are available to 2014). We find that melting in the northeastern AP, the focus area of this study, can be initiated both by sporadic westerly föhn flow over the AP mountains and by northerly winds advecting warm air from lower latitudes. A comparison of MAR with satellite and automatic weather station (AWS) data reveals that satellite estimates show greater melt frequency, a larger melt extent, and a quicker expansion to peak melt extent than MAR in the center and east of the Larsen C ice shelf. These differences are reduced in the north and west of the ice shelf, where the comparison with satellite data suggests that MAR is accurately capturing melt produced by warm westerly winds. MAR shows an overall warm bias and a cool bias at temperatures above 0°C as well as fewer warm, strong westerly winds than reported by AWS stations located on the eastern edge of the Larsen C ice shelf, suggesting that the underestimation of melt in this region may be the product of limited eastward flow. At higher resolutions (5km), MAR shows a further increase in wind biases and a decrease in meltwater production. We conclude that non-hydrostatic models at spatial resolutions better than 5km are needed to better-resolve the effects of föhn winds on the eastern edges of the Larsen C ice shelf.

## 1 Introduction

Increased meltwater production over the Antarctic Peninsula (AP) in the latter half of the 20[th] century has been linked to a warming atmosphere, with potential implications for future sea-level rise (Barrand et al., 2013; Turner et al., 2005; Vaughan, 2006). Surface melting has been implicated in the weakening and eventual collapse of ice shelves as well as the subsequent acceleration of contributing glaciers, with the Larsen A (1995) and Larsen B (2002) on the eastern

AP as the most notable examples (Vaughan et al, 1996; Rott et al, 1998; Scambos, 2004). In July 2017, a rift on the Larsen C Ice Shelf, which had been expanding for several years, resulted in the calving of the 5800 km$^2$ iceberg A68 (Hogg and Godmundsson, 2017).

Surface melting influences ice shelf stability through the stress produced by meltwater ponding as well as meltwater percolation through firn. One proposed mechanism for the disintegration of ice shelves hypothesizes that surface meltwater infills and deepens pre-existing crevasses, through a process called hydrofracture (Scambos et al., 2000; Weertman, 1973; van der Veen et al., 1997). In addition, a complementary mechanism proposes that when supraglacial lakes drain (becoming dolines), an upward flexure is induced which can weaken an ice shelf, both at the surface and at the base (MacAyeal and Sergienko, 2013). Large open-rift systems were observed over the Larsen B ice shelf in the summer of 2002 which are consistent with substantial melt initiating both mechanisms and leading to ice shelf disintegration (Glasser et al.,2008; MacAyeal and Sergienko, 2013). Alternatively, meltwater can affect ice shelf dynamics by percolating into firn and increasing its density until no pore space remains. In the absence of pore space, meltwater moves through the underlying ice sheet or collects on the surface in melt ponds. This process, operating over decades, can pre-condition the ice shelf for both hydrofracture and post-drainage flexure stress during high-melt seasons (Kuipers Munneke et al., 2014). Meltwater can also form below the surface in blue ice areas, due to the smaller extinction coefficients and lowered albedo of ice (Brandt and Warren, 1993), as well as under low-density snow on clear days, when temperatures are slightly below freezing (Koh and Jordan, 1995). Modeling studies suggest that the different sensitivities of subsurface blue-ice vs subsurface snow melt is a product of the radiative and heat transfer interactions resulting from their differing albedo, grain size and density (Liston et al,, 1999a; Liston et al.1999b). Meltwater forming over blue ice and flowing downstream to collect in subsurface layers (the ice-albedo feedback) has recently been shown to be substantial in parts of East Antarctica (Lenaerts et al., 2016). Recent work has also shown the lateral flow of meltwater (supraglacial runoff) on the Larsen A Ice Shelf in 1979 (Kingslake et al., 2017), which imply prolonged periods of lowered albedo. These surface rivers could become much more prevalent across Antarctica in future warming scenarios than previously expected, and may provide a means of stabilizing ice shelves by routing meltwater away (Bell et al., 2017).

Since the collapse of Larsen A and Larsen B ice shelves, ice velocities of several of their feeding glaciers have increased, and seasonal variations in flow have suggested that both summer meltwater percolation (Zwally, 2002) and the removal of backstress played a role in the acceleration (Scambos, 2004; Rott et al., 2002). The remaining Larsen C ice shelf to the south could prove to be similarly vulnerable to collapse due to atmospheric warming (Morris and Vaughan, 2013). Radar analysis over a 15-year period has shown that the surface of Larsen C has been lowering from both firn air depletion (due to either limited accumulation or high surface melt) and basal ice loss, although the latter term is thought to be more substantial (Holland, 2015). While most regional climate models (RCMs) do not account for englacial flow or surface rivers, accurate modelling of surface meltwater production is a crucial step in assessing the potential effects on the ice sheet, especially in the case of the Larsen C ice shelf.

The eastern AP, where the Larsen C ice shelf is located, is on average 3-5°C cooler than the western AP at the same latitude (Morris and Vaughan, 2013). When strong westerly winds force air across the bisecting mountain range of the AP (Fig. 1), the resulting föhn winds can produce pulses of warming on the eastern AP ice shelves

(Marshall, 2007). Föhn is a warm, dry air flow on the lee slopes of a mountain range (Beran 1967) that can contribute to melt and sublimation. Multiple studies have focused on the use of high-resolution non-hydrostatic models over the eastern AP to determine the frequency of föhn occurrence over relatively short periods (Elvidge et al., 2015; Grosvenor et al., 2014; Elvidge et al., 2016; King et al., 2017). King et al. (2017) found that over a single season, föhn flow occurred 20% of the time. This study showed substantial melt occurrence observed by satellites without föhn flow, suggesting that surface melt was influenced by other factors as well. A recent study by Turton et al., (2017), using a non-hydrostatic model, compared modelled flow characteristics during two föhn events and found that a 1.5 km version of the model was able to capture the eastward propagation of melt-inducing winds, whereas a 5km version could not, according to a comparison with AWS stations. However, Bozkurt et al. (2018) demonstrate that a 2km version of the same model was still unable to resolve high temperatures associated with the initiation of föhn flow during a short period. We note that because these modelling studies use a non-hydrostatic model, they are limited to short periods due to the prohibitive computational cost.

Models are limited by the parameterization of physics and our incomplete understanding of the physical processes driving the observed changes. Regional climate models (RCMs) such as the Modèle Atmosphérique Régionale (MAR), evaluated here, can be used for simulating the coupled atmosphere/surface system at a continental and decadal scale (Gallée and Schayes, 1994). The trade-off, in this case, is that RCMs might not be able to capture physical processes with the required accuracy and must be thoroughly evaluated with *in-situ* and remotely sensed observations. Several studies have used passive microwave estimates for melt occurrence alongside *in- situ* temperature data (Liu et al., 2003; Ridley, 1993; Tedesco et al., 2007; Tedesco et al 2009; Tedesco and Monaghan, 2009), reporting an increase of surface melting over the AP over the 1980-1999 period (Torinesi et al., 2003). However, other studies have suggested that the findings may have been impacted by a change in the acquisition hours of the satellite and that changes in melt over the 1979-2010 period were insignificant (Kuipers Munneke et al., 2012). Melt occurrence over the AP has also been investigated using the QuikSCAT satellite product at a ~2.225km resolution (Long and Hicks, 2010) in combination with model outputs from the RCM RACMO2 (Regional Atmospheric Climate Model) and *in-situ* temperature estimates (Barrand et al., 2013). Raw backscatter values from QuikSCAT have also been used to estimates melt flux over the AP (Trusel et al., 2013; Trusel et al., 2012). A recent study using 5.5km horizontal resolution run of RACMO 2.3 over the AP suggested that a further increase in resolution would be required to properly resolve föhn wind propagation, which would imply the removal of the hydrostatic assumption (Van Wessem et al., 2015a; Van Wessem et al., 2015b). However, Wiesenekker et al. (2018) show that föhn events observed by an AWS close to the AP mountain range were well captured by a later version of the same RCM, enabling a reconstruction back to 1979. Where the hydrostatic assumption is preserved (such as with MAR), higher resolutions may inhibit flow in the model, resulting in limited eastward föhn flow in the eastern AP (Hubert Gallée, personal communication). Despite these drawbacks, the current class of hydrostatic RCMs which include relatively complete representations of the snow physics are useful tools to simulate the effect of surface melt on the snowpack over long timescales. Additionally, these high-resolution runs can easily be compared to, and potentially nested into, continental-scale runs of the same model.        Here, we assess the MAR model at a 10 km horizontal spatial resolution over the AP, where outputs are available over a relatively long time period (1999-2014, i.e. 15 years), using both satellite and *in-situ* data,

aggregating meltwater production to drainage systems (basins) as described by Zwally (2002). While previous studies
have evaluated how surface melt is modelled using satellite data, or evaluated the representation of the near-surface
atmosphere with automatic weather station (AWS) data, we use both sources in conjunction to understand MAR's
ability to simulate specific physical processes, i.e. to assess melt and temperature biases by wind direction. We first
report total meltwater production from MAR at the basin scale and compare mean annual meltwater production with
outputs from RACMO2.3p2 (Van Wessem et al.m 2018), another hydrostatic RCM run at a 5.5km resolution (Sect.
3.1). We evaluate surface melt occurrence from MAR at the sub-basin scale using satellite estimates and link melt
occurrence biases to temperature and wind biases at a point scale using AWS data. We compare meltwater occurrence
derived from two satellite sources, passive microwave "PMW" and QuikSCAT active microwave, with MAR outputs
over the AP (Section 3.2). We focus primarily on the NE basin in the East AP as it contains the former Larsen A,
Larsen B and current Larsen C Ice Shelf, where we define sub-regions based on high and low melt occurrence
estimated by PMW algorithms (Tedesco, 2009). We then compare climatologies of melt extent, as well as inter-annual
trends, from both passive and active microwave data with those computed from MAR outputs (Section 3.3). Because
melt on the Larsen C Ice Shelf can potentially be initiated by northwesterly föhn flow sourced from over the AP or
southwesterly flow through gaps in the mountain range (even at sub-zero temperatures), we compare melt occurrence
reported by satellite estimates vs MAR (coinciding with the 2000-2009 QuikSCAT period) partitioned by temperature
differences and wind direction at the location of the Larsen Ice Shelf AWS. Two additional stations (AWS14 and
AWS15 are used to examine the persistence and spatial distribution of wind biases from 2009 to 2014. (Section 4).
Because all three stations are located on the eastern side of the Larsen C Ice Shelf, this comparison can assess the
impact of limited eastward flow on temperature and melt occurrence. In light of the model biases found in this analysis
and the potential to correct them with an enhanced resolution model in the future, the discussion (Section 5) includes
a sensitivity test with MAR at multiple resolutions. This is performed to specifically assess the effects of increased
resolution on eastward flow and resultant surface melt. Table 1 lists abbreviations used throughout the text along with
sections in which the terms are introduced.
**2. Data and Methods**
This study takes a combined observational and modelling approach. The primary tool used to understand the coupled
atmosphere and snowpack is the MAR RCM. We employ *in-situ* data collected from 3 AWS stations to evaluate the
near-surface atmosphere biases in MAR as well as to assess inter-annual trends. While in-situ observations of 2m air
temperature are frequently treated as a proxy for melt (Braithwaite 1981), this method is most effective when the
energy budget is dominated by the turbulent sensible heat flux and incoming longwave radiation and does not capture
melt which can occur due to shortwave radiative forcing when air temperatures are below 0°C (Hock, 2005; Kuipers
Munneke et al., 2012). We also use observations from the QuikSCAT (QS) and SMMR (Scanning Microwave
Multichannel Radiometer, 1978-1987) / SSM/I (Special Sensor Microwave/Imager, 1987 - to date) satellites to
evaluate both melt occurrence and intensity in MAR.

## 2.1 Regional climate model outputs

The MAR RCM is a modular atmospheric model coupled to the Soil Ice Snow Vegetation Atmosphere Transfer scheme (SISVAT) surface model (De Ridder and Gallée, 1998), which includes the multi-layer snow model Crocus (Brun et al., 1999). MAR was originally implemented to simulate energy and mass balance processes over Terra Nova Bay, Antarctica (Gallée and Schayes, 1994). Within SISVAT, meltwater is calculated at the surface when the surface reaches the melting point in combination with a surplus of energy (a deficit results in refreezing). The presence of meltwater alters the snow characteristics (for example, the type and size of snowgrains) and percolation through the snowpack is determined through a tipping bucket method based on snow density. A diagram and description of the sequence of these specific processes in MAR is provided in Figure S1.

The model configuration primarily used in this study is MAR version 3.5.2, with 23 sigma layers from 200 hPa to the surface. This version has been used in multiple studies over Greenland; the specific updates to the physics from the original version of MAR as well as multiple uses of this model are described in detail in Fettweis et al. (2016). The fresh snow density scheme used here is a new MAR implementation specific to Antarctica which has been tested with *in-situ* observations (Agosta et al., 2018, in review) and discussed further in that study. Here, fresh snow density ($\rho$) is computed as a function of 10m wind speed (WS, m s$^{-1}$) and surface temperature (Ts, K) such that:

$$\rho = 149.2 + 6.84 \text{ WS} + 0.48 \text{ Ts} \tag{1}$$

with a lower boundary of 200 kg m$^{-3}$ and an upper boundary of 400 kg m$^{-3}$. This parameterization was tuned such that the density of the first 50cm of snow fits observations collected over the Antarctic ice sheet, although we note that no reliable measurements were available over the AP. The subsequent compaction of snow layers uses the formulation from Brun (1989).There are 30 snow/ice layers of variable thickness from the surface to a 20 m depth (below which ice is assumed present). Topography is interpolated from 1 km Bedmap2 (Fretwell et al., 2013 ; Green et al., 2016) to the MAR grid. The snowpack is initialized at 300 kg/m$^3$ at the surface and 600 km/m$^3$ at depth. Following 2 years of spinup, MAR results are independent of the initial conditions ; for these results, 5 years of spinup were run.

MAR outputs are generated at a horizontal spatial resolution of 10 km for the years between 1999 and 2014. The model domain includes the AP region between -79.5° and -56.9° latitude and -94.9° and -39.7° longitude.  Lateral boundary conditions are specified from the European Centre for Medium-Range Weather Forecasts (ECMWF), using the ERA-Interim reanalysis (Dee et al., 2011), which is also used for a direct comparison with AWS wind speed/direction. This is a single model domain with no nesting. We note that the ice (vs sea) mask used does not include the Larsen A or Larsen B Ice Shelf in order to preserve consistency for comparison between years (most of which post-date the collapse of these ice shelves). For the analysis of the effects of resolution on surface melt estimates presented in Section 5, we use three version of MAR v. 3.9. Relative to version 3.5.2, which is primarily used in this study as well as in Fettweis et al. (2017), the computational efficiency of MAR v3.9 has been improved such that increased resolution runs are potentially viable. The improvements in the physics include an increase in the lifetime of clouds, partly correcting for the underestimation of downward longwave radiation and the overestimation of inland precipitation found in Fettweis et al. (2017). MAR v3.9 setups include a version at a 10km horizontal resolution similar to the model used for the main analysis, one where the horizontal resolution is reduced to 5km and one where the vertical discretization is increased to 32 sigma layers (at a 10km resolution).

We consider two conditions for identifying melting based on previous work comparing MAR outputs (version 3.2) and satellite microwave melt estimates that found that passive microwave estimates were sensitive to a meltwater content of 0.4% (or mm w.e.) in the first meter of the snowpack (Tedesco et al., 2007). The first condition ($LWC_{0.4}$) determines melt occurrence in MAR when the daily-averaged integrated liquid water content (LWC) in the first meter of the snowpack exceeds 0.4% for at least three consecutive days. The second condition ($MF_{0.4}$) determines melting when total meltwater production over the day exceeds 0.4 mm w.e., and is intended to capture both sporadic melt (which may refreeze) and melt which has percolated into the snowpack column below 1m, i.e. equivalent satellite-based estimates could have potentially shown melt occurrence during some portion of the day. A sensitivity test was conducted with multiple thresholds, finding that the differences between a threshold of 0.1 and 1 mm w.e. (suggested by Franco et al., 2013 as a melt threshold for Greenland) was negligible overall, but more substantial on the northern Larsen C Ice Shelf, where the 4 mm w.e. threshold proved insufficient to capture melt occurrence (Fig. S2). Similarly, we performed a comparison of melt occurrence computed from 2000-2009 at the Larsen Ice Shelf AWS for all satellite-based algorithms as well as AWS-based melt-occurrence criteria, i.e. where MaxT2m > 0°C and AvgT2m > 0°C (Fig. S2h). We found that neither total MAR melt occurrence nor the relative agreement with observed sources varied substantially between thresholds until a threshold of 4 mm w.e. Consequently, we use a meltwater production threshold of 0.4 mm w.e. to define melt occurrence for the remainder of the study due to its sensitivity at the northern Larsen C ice shelf. The differences in sensitivity for each satellite-based criteria for melt occurrences, as well as associated temperature biases, are discussed in detail in Section 3.1.

MAR meltwater production is compared to melt outputs from the RCM RACMO2.3p2, a hydrostatic model which has been run extensively over polar regions and over the AP at a 5.5km resolution at 40 vertical levels. RACMO2.3p2 is forced at the boundaries by ERA-Interim every six hours, as with MAR in this study. (Van Wessem et al., 2018). Model results over the AP for RACMO 2.3p2 did not vary substantially from RACMO2.3, which was evaluated extensively in previous work (Van Wessem et al., 2015a; Van Wessem et al., 2015b).

**2.2 Microwave satellite estimates of melt extent, duration**

Spaceborne microwave sensors can detect the presence of liquid water in snow over those regions where poor or no observations and unlike sensors in the visible range, microwave sensors are only weakly affected by the presence of clouds. In the case of active measurements (e.g., radar, scatterometer), the presence of wet snow is associated with a sharp decline in backscatter ($\sigma^0$) (Ashcraft and Long, 2000), whereas in the case of passive microwave data the detection is associated with an increase in brightness temperature ($T_b$) (Mote et al., 1993; Tedesco et al., 2007). In either passive or active microwave estimates, even the presence of a relatively small amount of liquid water (i.e. a few percent) triggers a substantial increase in the imaginary part of the dielectric constant (Ashcraft and Long, 2006; Ulaby and Stiles, 1980).

**2.2.1 Active Microwave Data: QuikSCAT**

We employ a wet snow high-resolution product (~2.225km) described in Steiner and Tedesco (2014) to derive melt occurrence from active microwave data. Both melt occurrence and raw backscatter values used in this analysis use

normalized backscattering values as measured by the Seawinds sensor onboard the QuikSCAT satellite at Ku band
(13.4 GHz), with the enhanced resolution provided by the application of the Scatterometer Image Reconstruction
(SIR) algorithm (Long and Hicks, 2010). Both Ku- and C-band scatterometers have been used extensively to detect
melt onset and freeze-up in Antarctica and Greenland (Drinkwater and Liu, 2000; Steiner and Tedesco, 2014; Ashcraft
and Long, 2006; Kunz and Long, 2006).

6       Threshold-based approaches with active microwave data, as used in this study, identify the point of melt

onset based on the departure in $\sigma^0$ from values in dry-snow with various thresholds (Ashcraft and Long, 2000; Ashcraft
and Long, 2006; Trusel et al., 2012). The approach used here derives melt occurrence from a threshold-based method
(ft3), which identifies melt when backscatter falls 3 dB below the preceding winter mean (Steiner and Tedesco, 2014;
Ashcraft and Long, 2006). This method, along with a wavelet approach have been evaluated over the AP with AWS
data at 5 locations; melt was assumed to occur at the AWS location when 2m air temperature exceeded 0°C for more
than 6 hours (Steiner and Tedesco, 2014).

13       In addition to the binary detection of melt, several methods have been proposed which relate seasonally-

integrated backscatter reduction to measures for melt intensity (Wismann, 2000;Smith, 2003; Trusel et al., 2013). As
these methods provide seasonally-cummulative values, we do not employ them in this study, although we do examine
raw backscatter values as a proxy for melt flux.
**2.2.2 Passive Microwave Data**
We complement the assessment of MAR with estimates of melt extent and duration obtained from passive microwave
observations which have been used in the past to assess melt occurrence in Antarctica and Greenland using brightness
temperature at 19.35 GHz with a horizontal polarization (Tedesco, 2007). One of the major disadvantages of passive
microwave is the relatively coarse horizontal spatial resolution (25 km) with respect to the fine-scale topography
characterizing the AP. However, the historical record for passive microwave data extends as far back as 1972.
Threshold-based methods for melt detection from passive microwave data range from a combination of multiple
frequencies and polarizations (Abdalati and Steffen, 1995) to using a single frequency, single polarization (e.g., Mote
et al. 1993, Tedesco 2009), as is used in this study. Three algorithms are used here which are described in detail in
(Tedesco, 2009). These include the *240*-algorithm where the threshold was determined as the value above which an
increase in liquid water content above 1% no longer produces an increase in $T_b$, based on output of an electromagnetic
model. The original threshold of 245K was found to be insufficiently sensitive and reduced to 240K for this study
(Tedesco, 2007) (M. Tedesco, personal communication). The second algorithm uses the winter mean threshold-based
method ALA:
$\quad T_c = T_{winter} * \alpha + T_{wet_{snow}} * (1 - \alpha)$                     (2)
where snowmelt is assumed to occur when the brightness temperature ($T_b$) exceeds a threshold brightness temperature
($T_c$) based on the mean winter (JJA) $T_b$, the wet snow $T_b$ ($T_{wet\_snow}$, equal to 273K) and a mixing coefficient ($\alpha$, equal
to 0.47). For the ALA algorithm, Ashcraft and Long (2006) presume a wet layer of 4.7 cm and a Liquid Water Content
of 1%. Finally, the third algorithm (zwa), determines melt occurrence when $T_b$ exceeds a threshold value $T_c$ which is
based on the on the winter mean threshold ($T_{winter}$) and a threshold value ($\Delta T$), in this case 30K (Zwally and Fiegles,

2     1994)

$T_c = T_{winter} + \Delta T$                                                              (3)
**2.3 AWS measurements**
We evaluate the MAR simulation of the near-surface atmosphere using pressure, temperature and wind speed data
collected by three automatic weather stations (AWS) on the AP (Fig 1). The comparison between MAR outputs and
AWS data for surface pressure are provided in supplementary data. Data from the Larsen Ice Shelf AWS is obtained
from the University of Wisconsin Madison (AMRC, SSEC, UW-Madison) at a 3-hourly temporal resolution. AWS
data from two additional sites on the Larsen Ice Shelf (AWS14 and AWS15) are obtained from the Institute for Marine
and Atmospheric Research at Utrecht University (IMAU) at an hourly resolution (Kuipers Munneke et al., 2012). We
note that the Larsen Ice Shelf AWS (-67.00 °S, -61.60°W) and AWS14 (-67.00°S, -61.5°W) fall within the same MAR
grid cell.

13         AWS values are temporally averaged to obtain mean daily values for the comparison with MAR outputs.

Metrics are computed for December-January-February (DJF, summer). We did not compute a seasonal average when
more than 5 consecutive days of data were missing. The five-day period was chosen as an upper limit for the length
of a synoptic event, corresponding spatially to approximately 145 MAR grid cells (or half the model domain) of
continuous flow in a single direction for an average windspeed of 3.4 m/s, which is the expected value (i.e. the
predicted mean based on the Weibull distribution), for Larsen Ice Shelf AWS in DJF from 1999-2014 (Fig. 7c). Near-
surface (2m) air temperature values are corrected for a difference between AWS station elevation and the elevation
averaged by the corresponding MAR gridcell by calculating the elevation gradient from surrounding MAR gridcells
and interpolating the final value for the AWS location's recorded elevation using the Bedmap2 DEM (Fretwell et al.,
2013). Differences in elevation values between MAR at 10km resolution and those recorded at AWS stations were as
large as 23 m. Maximum daily 2m air temperature (MaxT2m) is calculated as well because this measure may help
capture sporadic melt events. MaxT2m values are extracted from available 3-hourly values and are used only when
no more than one 3-hour measurement is missing during the day. Pressure values from AWS stations are also observed
at approximately 2m above the surface, and compared to MAR values at the first atmospheric layer in MAR. Because
the height of this layer is generally between 2 and 3 m above the surface, this is treated as an acceptable proxy for 2m
pressure values. Pressure values from MAR are corrected for elevation using the hypsometric equation (Wallace and
Hobbs, 1977).
**2.4 Statistical Methods**
To evaluate and quantify the differences between MAR outputs and AWS data for temperature and wind speed we
use a mean bias. Additional statistical measures shown in supplemental data include the coefficient of determination
($R^2$), root mean squared error (RMSE) and mean error (ME) (Wilks, 1995). We assess the extent to which each station
is representative of larger scale climate variability by constructing correlation ($R^2$) maps between MAR values co-
located with AWS stations vs all other gridpoints in the full MAR domain (Fig. S7). We ignore all $R^2$ statistics where
the p-value exceeds 0.05.
To capture wind speed frequency distributions, we fit available data for each season for MAR (for the full
2000-2009 period), AWS (when AWS data are available) and MAR-R (MAR values collected only when AWS data
is available) with a Weibull distribution (Wilks, 1995). The shape (β) parameter roughly captures the degree of skew,
with higher values being closer to a normal distribution. The scale (λ) parameter approximates the peak frequency
(we note that this is not equivalent to the arithmetic mean). We report expected values (i.e. first moment or mean) for
each windspeed distribution using the best Weibull fit.
**3. Results: Melt Occurrence and Meltwater Production**
In this section, we show results concerning total meltwater production in the AP and compare melt occurrence
estimated by MAR with estimates from three passive microwave algorithms as well as QuikSCAT ft3. The relative
sensitivity of each melt occurrence criteria, as well as their associated temperature biases, are first compared at the
location of the Larsen Ice Shelf AWS. We then identify spatial biases for melt occurrence at the domain scale, finding
substantial differences in the center of the Larsen C Ice Shelf as well as to the north and west of the NE basin, a region
which includes the former Larsen A and B ice shelves as well as the northernmost portions of the Larsen C ice shelf
(Section. 3.2). These differences could result from either weaknesses in the MAR representation of wind dynamics
(discussed in Section 4) or from limitations of the satellite sensor or algorithm. Finally, we compare the climatology
and inter-annual variability of melt extent (calculated by multiple algorithms) over the CL and NL region (Section.

19 3.3).

**3.1 Meltwater production over the AP**
We show MAR meltwater production over the 1999-2009 period (Fig. 2). The total annual meltwater production
estimated by MAR shows substantial inter-annual variation with the NE basin accounting for the highest meltwater
production, closely followed by the SW basin (in green).  The NE basin is divided into three regions: the NL and CL
masks (discussed in Section 3.2) and the remainder of the basin. We note that the SW basin does not covary with the
NE basin and the subregions of the NE basin do not consistently covary with one another. The meltwater production
shown here does not account for refreezing and we note that the effects of refrozen melt on the snowpack will vary
regionally depending on local properties. The NL region dominates meltwater production in the NE basin in most
years except for 1999-2000, 2002-2003 and 2003-2004. The 2001-2002 melt season shows the second lowest overall
melt production during the study period (only the preceding year is lower). Declining aggregate meltwater production
across the AP does not necessarily correspond to declining meltwater production in the most vulnerable regions of the
northeastern AP (including the Larsen C Ice Shelf). Because melt in the NL region is particularly sensitive to föhn-
induced melt, we note that changes in circulation patterns may affect the northwest regions differently than the
southern regions. The strong relationship between wind direction and temperature bias points to the need for isolating
dominant inter-annual patterns of melt in the Northern Larsen C Ice Shelf and associating them with large-scale
atmospheric drivers.

3        A comparison between mean annual meltwater production from 2000-2009 calculated using RACMO2.3p2

(5.5 km) vs MAR (10km) is shown in Fig. 3. MAR shows higher meltwater production overall (Fig. 3b vs 3a), with a
difference of over 150 mm w.e. on the Larsen C ice shelf north of 67°S latitude. Over the NE basin, MAR meltwater
shows enhanced meltwater production near the AP mountains, including towards the southern edges, and declines
eastward and southward. By comparison, meltwater production from RACMO2.3p2 melt declines southward, but no
similar west-to-east gradient is apparent. Although inter-annual standard deviations over the northern Larsen C ice
shelf are generally above 100 mm w.e. in both models, there are major differences in other regions, with MAR
meltwater production exceeding RACMO2.3p2 values by 30 mm w.e.on the southern Larsen C ice shelf as well as
the George VI ice shelf (Fig. 3d vs 3c). Van Wessem et al. (2015a) suggest that even at 5.5 km resolution, the
underestimation of the height and slope of the orographic barrier may result in an underestimation of föhn winds as
well as precipitation in RACMO2.3p2. We note that in addition to the difference in horizontal model resolution,
RACMO2.3p2 contains 40 atmospheric layers while MAR implements 23 layers. While the differences in total
meltwater production from RACMO2.3p2 and MAR could be a product of dissimilar physics, the potential effect of
model resolution on meltwater production in MAR is specifically discussed in Section 5. While melt occurrence and
meltwater production are not related in any linear fashion, we note that the spatial pattern produced by MAR, i.e. the
eastward gradient from the edge of the AP, is also shown in observed melt occurrence estimates, most notably from
the PMW zwa and QS algorithms (Fig. 5f,g), as discussed in greater detail in the next section.
**3.2 Melt occurrence over the AP**
Fig. 4 shows melt occurrence (in days) at the Larsen Ice Shelf AWS location (shown in Fig. 1) as estimated from the
satellite-based algorithms QuikSCAT ft3 (Section 2.2.1), three passive microwave algorithms (Section 2.2.2),
temperature-based criteria from the AWS station (MaxT2m > 0°C and AvgT2m > 0°C), and the $MF_{0.4}$ metric derived
from MAR (Section 2.1). At this location, we find that QuickSCAT ft3 and PMW ZWA show the greatest sensitivity
to melt occurrence. Of the AWS-based metrics, M (MaxT2m > 0°C) shows a sensitivity to melt occurrence comparable
to PMW ALA while the T metric (AvgT2m > 0°C) compares poorly to satellite-based measures (Fig. 4a). We find
that at colder temperatures (when MAXT2m < 0°C), AvgT2m values reported by MAR are substantially higher than
those reported by the AWS when only MAR reports melt (Fig. 4b). However, at higher temperatures (where MaxT2m
>= 0°C), the AWS reports higher MaxT2m temperatures than MAR and biases are even stronger when only
observation-based metrics report melt (Fig. 4e). We note that the Larsen Ice Shelf AWS is located on the eastern edge
of the Larsen C ice shelf and the major discrepancies in melt occurrence at this location will be explored further in
Section 4, where we further expand the analysis of melt occurrence and temperature biases to include wind direction
biases as well.

34        In Fig. 5, we show melt occurrence over the full domain derived from satellite sources, both metrics derived

from MAR (Section 2.1) as well as the $MF_{0.4}$ criteria applied to RACMO2.3p2. QuikSCAT ft3 generally estimates
higher average yearly melt occurrence than either of the MAR melt metrics over the full domain. In the NE basin, the

difference is on the order of 25 more days than the MAR $MF_{0.4}$ melt metric (Fig. 5g). Differences between QuikSCAT ft3 and $MF_{0.4}$ also show a strong latitudinal dependence in the NE basin, shifting from near agreement in the northern regions of the Larsen C Ice Shelf to QuikSCAT ft3 reporting over 500% of the melt days reported by MAR towards the southern edge. Melt onset is on the order of 22 days earlier in QuikSCAT ft3 than in $MF_{0.4}$ in the NE basin, except at the northern edge of the Larsen C ice shelf, where $MF_{0.4}$ reports average yearly melt onset as much as 25 days earlier than QuikSCAT ft3 (Fig. S3). A comparison between the two MAR melt metrics shows that $MF_{0.4}$ reports as much as 40 more days of melt than $LWC_{0.4}$ at the northern tip of the Larsen C Ice Shelf (Fig 5b vs Fig 5a). The portion of the Larsen C ice shelf which experiences an average of 25 days of melt or more extends as far south as 80.0°S on the eastern side of the Larsen C ice shelf according to the $MF_{0.4}$ metric but extends only to 70.5°S according to $LWC_{0.4}$. Towards the very south of the Larsen C Ice Shelf, the two MAR metrics show similar values, although $LWC_{0.4}$ reports melt onset as late as early January (Fig. S3a) while $MF_{0.4}$ reports melt onset in December (Fig S3b). The formulation for the $MF_{0.4}$ metric, which considers melt at any time of the day for the full depth of the snowpack, suggests that the early season melt observed only by $MF_{0.4}$ is either sporadic (i.e. can refreeze) and/or percolates below 1m in the snowpack in the south of the Larsen C Ice Shelf, i.e. below the depth range at which $LWC_{0.4}$ is calculated. Whereas QuikSCAT ft3 and MAR melt metrics report maximum melt occurrence in the north and west of the Larsen C Ice Shelf ($MF_{0.4}$ reporting > 60 days, Fig. 5b), PMW algorithms report maximum melt occurrence in the center-east of the Larsen C Ice Shelf, specifically 43 days (240, Fig. 5c), 57 days (ALA, Fig. 5d) and 69 days (ZWA, Fig. 5e). RACMO2.3p2 reports substantially higher melt occurrence than MAR at the center of the Larsen C ice shelf as well as a comparatively limited west to east gradient. Because overall average annual meltwater production in MAR was shown to be substantially higher, with a stronger west-to-east gradient away from the AP (Fig. 3), we conclude that in comparison to RACMO2.3p2, MAR produces melt less frequently, but with greater intensity.

In summary, a comparison between observed and modeled data sources show two distinct spatial patterns for maximum melt occurrence. QuikSCAT ft3 as well as both MAR melt metrics show the highest range of melt days in the northern and western edges of the Larsen C Ice Shelf (including both high and low elevation regions) while PMW algorithms show the highest number of melt days in the center of the Larsen C Ice Shelf, where elevations are lower and topography is less complex. We hypothesize that the major difference in spatial patterns between algorithms/melt metrics is related to the different resolutions of the data sources (~2.2225 km for QuikSCAT, 10km for MAR and 25km for PMW), such that QuikSCAT is better able to resolve melt where topography is complex , such as near the spine of the AP. Secondarily, the differences are a product of the depths presumed for the calculation of meltwater content. This is true for both the MAR metrics and for the three PMW algorithms; the "ALA" algorithm, for example, presumes a 4.7cm depth and a 1% liquid water content. (see Section 2). To confirm this, we find the maximum depth to which meltwater percolates (according to MAR) associated with the number of days when melt occurs (according to PMW algorithms). Histograms for total PMW melt days in Fig. S4 show three peaks (two major inflection points) for each algorithm which are used to create three classes for meltwater occurrence ("low", "medium" and "high"). For these classes, the maximum depth to which meltwater percolates (in MAR) is shown in Fig. S6 and the associated elevation and MAR meltwater production is shown in Table S1.

Spatial regions defined as having "low" melt occurrence are highly heterogeneous with regard to elevation,
meltwater percolation and the relative sensitivity of PMW algorithms. Low melt occurrence regions largely include
the spine of the AP and regions just east of it. Bedmap2 (Fretwell et al., 2013) reports a large range of elevations while
MAR reports low coincident meltwater production and a relatively shallow meltwater depth. Both the ALA and ZWA
algorithms report melt at higher elevations (above approximately 1300m and 1900m, respectively) than the 240
algorithm, which neither reports any melt occurrence above 1100m in the NE basin nor at lower elevations to the north
and south. (Table S1, rows 1,4,7 and Fig. S6). Where melt occurrence is low, the 240 and ALA algorithms generally
detect melt only where MAR reports a maximum meltwater percolation depth below 0.4 m, (Fig S6a,b), whereas the
ZWA algorithm can detect melt at a substantially shallower depth of 0.1 m (Fig S6c). Although generally meltwater
in MAR rarely percolates below 3m, in low melt-occurrence regions, modeled meltwater occasionally percolates
below 10m in the beginning of the melt season (Fig. S6 a,b,c, column "N", indicating November). We remind the
reader that melt occurrence within the firn layer (as calculated by MAR $MF_{0.4}$) will capture melt that can refreeze
immediately, so this does not necessarily correspond to melt which is retained in the snowpack. Rather, the snowpack
layer depth represents the deepest layer which is affected by the melt process according to MAR.
By contrast, where PMW reports high melt occurrence in the NE basin, MAR consistently reports high
coincident meltwater production, low elevations and the deepest average meltwater percolation in the region. In the
month of January, we find that where PMW algorithms report melt, coincident MAR meltwater percolates to 2 m into
the snowpack for 35-47% of the total day-pixels in the NE basin which report any melt, and as deep as 3 meters for
more than 30% of total day-pixels  (Table S1, 240-H, ALA-H, ZWA-H, Fig. S6 g,h,i).
To quantify the two major spatial trends for maximum melt occurrence, i.e. (1) PMW in the center-east of
the Larsen C ice shelf and (2) QuikSCAT ft3 and MAR in the northwest of the NE basin, we (a) explicitly calculate
concurrent melt occurrence in all PMW algorithms (PMWAll) for the first region and (b) define the latter
geographically in order to include most of the NE basin, but deliberately exclude center-east of the Larsen C ice shelf
region where PMW melt is highest. The first region "CL" (Center Larsen, as the entire region is restricted to the Larsen
C ice shelf), where all PMW algorithms agree on high melt occurrence, is defined where PMWAll reports average
yearly total melt days exceeding one standard deviation from the mean for the NE basin. Mean elevation for the CL
region is 42.70±17.70σ m (where σ is one standard deviation). PMWAll reports a mean annual 36 days of melt
occurrence (vs 21 days derived from $MF_{0.4}$) and the mean annual MAR meltwater production calculated only where
PMWAll reports melt occurrence is 96 mm w.e./100km$^2$ (vs 143  mm w.e./100km$^2$ when $MF_{0.4}$ reports melt)(Table
S1, row 11,12).
The "NL" (Northern Larsen) mask is defined by finding the mean latitude of the CL region and including all
portions of the NE basin above this latitude, but excluding the CL region (Fig. 2, inset). In the NL region, elevation is
highly-variable, with a mean value ~600m and MAR and QS detect melt both earlier and more often than for PMW
algorithms. The NL region includes the eastern spine of the AP and most inlets (including Cabinet Inlet and SCAR
Inlet), a small portion of the northern Larsen C ice shelf and all regions surrounding the former Larsen A and Larsen
B ice shelves.

**3.3 Climatology and inter-annual trends for melt extent at the sub-basin scale**

We compare the seasonal cycle and interannual variability of melt as modeled by MAR vs observations for both the CL and NL regions by computing regional melt extent over the 2000-2009 period (total melt extent area for each day in NDJF), for each year as well as the climatological average. The PMWAll algorithm is typically treated as the most restrictive condition while the PMW zwa and QuikSCAT ft3 are the most sensitive. Melt extent is defined as the total area reporting melt daily between Nov 1st and February 28th (austral summer, including November to show early melt)(Fig. 6).

The melt extent climatology for PMWAll in the CL region shows the initial increase in sustained melt occurring around December 15th with melt extent peaking in January, followed by a series of increasingly smaller melt pulses ending with refreezing at the end of February. While MAR shows peak melt extent at the same point in the season, the progression from melt onset is more gradual, average peak melt extent is generally smaller and interannual variability (indicated by the grey envelope) during peak melt extent is larger (Fig. 6c vs Fig. 6a). In the CL region, the PMWAll metric is generally restricted by the low sensitivity of the 240 algorithm. Interannual variability for melt extent is substantial, with PMWAll reporting a larger melt extent than MAR towards the middle of the melt season in most years (Fig. 6b,d), but not necessarily during melt onset or its ending. In the CL region, PMWAll reports a larger melt extent throughout the melt season during 2000-2001 and 2001-2002 (Fig. 6d). During three periods, MAR reports a larger melt extent than PMWAll, including 1999-2000, the latter half of the 2002-2003 season and the 2003-2004 season. While the highly-sensitive PMW ZWA algorithm (Fig. 6e,f) reports sporadic periods where MAR melt extent is larger (during the 1999-2000 and 2003-2004 melt seasons, for example), ZWA generally reports either a larger melt extent or general agreement with MAR. Similarly, melt extent derived from the QuikSCAT ft3 algorithm consistently shows a larger melt extent than MAR, except for a few short periods towards the end of the season in 1999-2000 and 2003-2004 (Fig. 6 g,h). We note that for several years, both QuikSCAT ft3 and PMW ZWA report substantial melt occurrence early in the season (~Nov 15th) and that the QuikSCAT ft3 climatology frequently reports melt occurrence in the CL region well after February (Fig. 6g).

The NL region includes areas which reported low melt occurrence in all PMW algorithms, variable meltwater percolation depth in MAR was variable , and a large range of elevations was observed (Section 3.2), implying that the mask defined by the combined PMWAll algorithm is less clearly linked to consistent modeled physical properties in this region. Here, the MAR melt extent climatology (Fig. 7a,b) is consistently larger than PMWAll throughout the season (Fig. 7c,d). In comparison to the ZWA (Fig. 6c) and QuikSCAT ft3 (Fig. 7g) algorithms, MAR reports less melt extent in the middle of the season (with peak melt extent in January), but larger melt extent at the beginning and end of the melt season. As compared with the CL region, the MAR climatological melt extent shows less inter-annual variability (grey envelope, Fig. 7a). During the 2000-2001 and 2001-2002 melt seasons, MAR shows a larger melt extent than PMWAll (Fig. 7d), but less than the PMW ZWA (Fig. 7f) or QuikSCAT ft3 (Fig. 7h) algorithms. We find that during the 2005-2006 season, MAR shows greater melt extent than PMWAll, consistently less than QuikSCAT ft3, but reports a greater melt extent than ZWA only towards the end of the season. We consider the condition where only QuikSCAT ft3 or PMW ZWA show a greater melt extent than MAR to be potentially indicative of sporadic surface melt.

In summary, we conclude that in the CL region, MAR reports a larger melt extent from 2009-2009 than
PMWAll (which is highly-restrictive), but a smaller melt extent than either the PMW ZWA or QuikSCAT ft3
algorithms, which are more sensitive. Notably, MAR melt occurrence is comparatively low during the peak melt
period. By contrast, in the NL region, MAR reports greater melt occurrence than the most restrictive measure
(PMWAll) during peak melt, but far less than the highly-sensitive QuikSCAT ft3 algorithm. The interannual
comparison suggests that MAR shows substantially less melt occurrence than observations during the 2000-2001 and
2001-2002 seasons in the CL region, but not the NL region.
**4. Results: Wind and Temperature Biases at the Larsen Ice Shelf station**
The eastern AP is generally substantially colder than the western AP, and temperature-driven melt primarily results
from either large-scale advection from lower latitudes or from westerly föhn flow over the spine of the AP (Marshall
et al., 2006). Here, we assess the bias in temperature and melt occurrence associated with wind direction at three AWS
locations on the Larsen C Ice Shelf (shown in Fig. 1). We first discuss wind direction and wind speed biases during
the summer season at all three locations (without regard to melt occurrence) (Section 4.1). For prominent wind
direction biases, we quantify the associated temperature and melt occurrence biases in order to capture atmospheric
conditions where MAR reports less melt occurrence than observations (Section 4.2). All MAR and satellite data used
are co-located to the grid cell associated with the AWS (Fig. 1), and we remind the reader that all three stations, at the
eastern edge of the CL region (Fig. 2 inset), are located where MAR reported substantially less melt occurrence than
PMW algorithms, QuikSCAT ft3 or AWS temperature-based criteria.
**4.1 Aggregate wind direction biases**
Fig. 8 shows wind frequency distributions during the summer season, color-coded for wind direction as represented
by the pie graph at the right. We note that AWS data are 3-hourly averages and ERA-Interim are 6-hourly averages
for wind speed and direction, while MAR produces daily-averaged outputs. For this reason, a direct comparison
between Weibull parameters derived from MAR vs AWS data is not fully justified. The Larsen Ice Shelf AWS has
full temporal coverage during the QuikSCAT period while AWS14 and AWS15 were installed after termination of
the QuikSCAT mission. These last two stations are used in this study to demonstrate that (a) similar wind biases
persisted after the QuikSCAT period at multiple locations, as AWS 14 the Larsen Ice Shelf AWSs are co-located to
the same MAR grid cell and that (b) wind biases vary slightly by latitude, AWS15 being located slightly to the south.
Both MAR and AWSs at all stations show a larger proportion of northerly winds at lower windspeeds (Fig. 8, in
yellow and blue), although AWSs report a greater frequency of southwesterly and northwesterly flow (Table 2 col.
4,5 rows 4-9). At the Larsen Ice Shelf AWS location, both AWS and MAR report dominant northeasterly flow (Table
2, rows 4,8, col2). However, the Larsen Ice Shelf AWS reports slightly more flow which is either southwesterly
(28.9% for AWS vs. 23.2% in MAR) or northwesterly (19.3% for AWS vs. 14.1% in MAR) while MAR reports more
southeasterly flow overall (23.5% in MAR vs. 17.4% in AWS). These biases are more pronounced at the southern
AWS15, where modelled temperature correlates with a larger portion of the southern Larsen C Ice Shelf than for
AWS14 (Fig. S7, Fig. 8i,j). ERA-Interim reports substantially more northwesterly flow than either AWS or MAR and

a smaller proportion of southwesterly flow in the 180°- 225° range (especially at the southernmost AWS15 location), although easterly flow is equivalent to AWS-reported estimates. We note that although ERA-Interim has been shown to reproduce the basic structure of föhn flow (Grosvenor et al., 2014), the horizontal spatial resolution may be too coarse to adequately capture southwesterly gap flow here. As discussed further in Section 5, westerly flow towards the stations used in this study may be strongly affected by the fine-scale representation of topography (which is coarse in ERA-Interim) and the lowered orographic barrier due to the smoothing of topography in the northwest in ERA-Interim may contribute to the enhanced northwesterly flow reported by ERA-Interim.

**4.2 Wind and temperature biases concurrent with observed melt occurrence**

When daily-averaged temperature (AvgT2m) values are high, it is more likely that melt is sustained, while high maximum daily temperatures (MaxT2m) can also occur during sporadic melt. Melt occurrence is strongly influenced by the temperature of the snow column as well as at the surface; internal melting can occur even when the surface is frozen due to net outgoing longwave radiation (Holmgren, 1971; Hock, 2005). It is therefore possible for melt to occur despite a cold bias. In general, we find a small, but consistent warm MAR bias for AvgT2m, and a consistent cold MaxT2m bias (Table 2, rows 12,13). However, when we restrict the dataset to days when AWS-recorded temperatures exceed 0°C, a condition where melt is most likely, MAR indicates a cold bias for AvgT2m and an enhanced cold bias for MaxT2m (Table 2, rows 15,16). This implies that MAR is colder than observations at the temperature ranges where melt is likely, although melt is still possible due to other components of the energy balance.

The cold MaxT2m temperature bias is strongest during northerly flow in general (Table 2, row 13,16, col 2,5), but strongest during easterly flow on the days when MAR reports melt (Table 2, row 23,26, col 2,3). Satellite-based melt is detected primarily when AWS-recorded flow is northeasterly (0°-90°) or southwesterly (180°-270°), with PMW(QS) reporting 42%(36%) northeasterly flow and 29%(26%) southwesterly flow. On days when MAR reports melt (Table 2, rows 19,20), southeasterly flow in MAR is more prominent (while AWS values decline) while the proportion of northwesterly flow declines (but increases at the AWS). We find that the major flow biases account for a relatively small proportion of melt which is captured by observations but not by MAR. The easterly flow bias accounts for 8%(9%) of days where PMWAll(QS) melt occurrence is not also captured by MAR (Table S9) while the southerly flow bias accounts for 6%(6%) of days when PMW(QS) melt occurrence is not also reported by MAR (Table S8). For these wind direction biases, Fig. 9 presents temperature values when observed sources, either PMW All or QuikSCAT ft3, report melt, but MAR does not. We refer to the condition where PMWAll reports melt (but MAR does not) as "PMWEx" (i.e. PMW exclusive-or), with the equivalent condition for QuikSCAT ft3 called "QSEx". We limit the melt days shown in each figure panel to a specific wind bias, thus showing how the wind bias directly influences temperature-driven melt in both satellite-based observations as well as MAR. Tables S8-S12 contain relative proportions of each case (flow bias) divided for each restriction (i.e. MAR, QSEx or PMWEx), as well as the timeseries mean and biases for AvgT2m, AvgT2m>0°C (excluding days when AvgT2m values from AWS are below 0°C), MaxT2m and MaxT2m >0°C.

For the main biases, i.e. when MAR either reports northerly winds as southerly (Fig. 9a,b) or westerly winds as easterly (Fig. 9 c,d), modelled temperature values are clustered around 0°C, whereas AWS-observed temperatures,

especially when only satellite-observed melt occurs, are higher. When MAR reports melt, MAR AvgT2m values cluster near 0°C, with a small overall warm bias (Tables S8,S9, row 4, col 8). Under omission conditions (PMWEx and QSEx), AvgT2m values are lower, and the MAR bias is slightly negative, although the standard deviation is high (Tables S8, S9, row 5,6, col 7). With all flow cases, only QuikSCAT ft3 shows melt at very low observed AvgT2m values. By contrast, AWS MaxT2m values are substantially higher than MAR values (the latter clustering around 0°C) (Fig. 9b,d). Temperature biases associated with southwesterly flow are similar to those shown by the overall bias towards easterly flow in MAR, and are shown in Table S10,S11.

Northwesterly winds are most likely to produce föhn-induced melt and we find that on days when MAR reports melt, only 13.2% of winds are northwesterly while AWS reports 25.2% of flow as northwesterly (Table 2, rows 9,10, col 5). Northwesterly winds show the highest expected windspeeds as well as the highest standard deviation for both MAR and AWS (Table 2, rows 19,20, col 5). While the temperature bias when wind directions are in agreement is relatively minimal, the temperature bias when northwesterly winds are misrepresented is substantial. When MAR reports melt but misrepresents northwesterly winds (this condition accounts for 3% of all MAR melt days), the cool bias for MaxT2m > 0°C is above 4°C (Table S12, row 4, col 10). For the PMWEx condition (when PMW reports melt but MAR does not), AWS MaxT2m values exceed MAR values by more than 5°C (Table S12, row 5, col 10). Despite the strength of the temperature bias, this wind direction bias accounts for only 3% of melt in MAR and only 3%(4%) of melt occurrence reported by PMWEx(QSEx). By contrast, when westerly flow is modelled accurately, MAR captures higher AvgT2m values, which frequently exceed 0°C, with a slight cool MAR bias when AvgT2m > 0°C (Fig. 9e). The PMWEx and QSEx conditions still report melt at lower temperature values, and the MAR bias remains positive. Although a cold MAR bias persists, MaxT2m values are generally in better agreement at the Larsen IS AWS location during this condition (Fig 9f, Table S12).

**5. Discussion and Conclusions**

We conclude that MAR captures melt which occurs just east of the AP (which is normally the product of westerly föhn flow) with acceptable accuracy according to satellite estimates, but that that melt is underestimated with respect to both AWS and satellite estimates in the eastern part of the Larsen C Ice Shelf. This is partially the result of limited westerly flow in MAR towards the eastern part of the Larsen C ice shelf, as compared to AWS estimates. Specifically, MAR shows lower melt occurrence than satellite estimates in the center and east of the Larsen C Ice Shelf (i.e. the CL region, where eastward flow is likely limited in MAR ), while in the north and west of the NE basin (i.e. the NL region which is most immediately affected by westerly flow), MAR reports melt occurrence largely concurrent with satellite estimates. The NL region fits a spatial pattern of föhn-induced melt just lee of the AP and extending eastward from inlets which has been shown in previous studies (Grosvenor et al., 2014) and particularly in the northernmost portion of the NE basin surrounding the Larsen B ice shelf, where the correlation between föhn winds and satellite-based melt occurrence has been shown to be as high as 0.5 between 1999-2002 (Cape et al., 2015, Fig. 12). For example, within the CL region, there are periods during the 2001-2002 season when MAR reports no meltwater production, but raw QuikSCAT backscatter values report periods where over 300 km$^2$ of surface area show backscatter values dipping below -15 dB (Fig. S9e).

MAR reports warmer temperature compared to AWS observations recorded on the east of the Larsen C ice shelf at temperatures below 0°C, when melt is less likely to occur, but which may still impact the refreezing process. However, when maximum daily temperatures (MaxT2m) and average daily temperatures (AvgT2m) exceed 0°C, MAR shows a substantial cold bias. This is particularly evident when MAR misrepresents westerly winds or northerly winds, and the temperature bias is most extreme when northwesterly flow is misrepresented, i.e. the condition when the most intense föhn flow would be likely. However, this represents only a small proportion of the melt occurrence bias, i.e. melt occurrence reported by satellite estimates, but not by MAR.

We demonstrate the impact of westerly winds on melt during a single season, specifically during both mid-December and the beginning of January of the 2001-2002 season. During both of these periods, satellite-based melt extent in the CL region increases substantially, while MAR melt extent declines after an initial pulse (Fig. S9a). In December, MAR shows an increase in northwesterly flow, both at the station and throughout the region while AWS reports northwesterly winds at slightly higher speeds. Beginning approximately on January 1[st], the NL region reports substantial northwesterly flow, followed by southwesterly flow, although neither is reported at the Larsen Ice Shelf AWS station east of the NL region. Over January, while both AWS and MAR report northeasterly flow, the AWS station also reports substantial high-speed southwesterly flow not captured by MAR. After this period (beginning on approximately Jan. 1[st]), AWS AvgT2m temperatures consistently exceed MAR AvgT2m values until the end of the season (Fig. S11), suggesting that because MAR did not accurately model the initial intrusion of westerly winds, subsequent temperature-induced melt was limited over the eastern Larsen C ice shelf, where this AWS is located. Presuming that the flow characteristics are largely similar in this relatively flat region, we conclude that the underestimation of melt in the CL region is partially due to the absence of westerly flow, but that this flow is adequately captured directly east of the AP (comprising the NL region).

Previous work has suggested that southwesterly föhn winds can result from gap flow (Elvidge et al. 2015), although we note that the southwesterly jets studied in this single campaign were typically cooler and moister than surrounding air, i.e. föhn flow produced from isentropic drawdown. While a version with a higher spatial resolution may potentially resolve topography sufficiently to include the initial intrusion of southwesterly gap flow, as well as northwesterly föhn flow, it may also further inhibit subsequent eastward flow when the hydrostatic assumption is retained. While a higher resolution of MAR v3.5.2 (used throughout this study) was not run due to computational constraints, the enhanced computational efficiency of a newer version of the MAR model (MAR v3.9, Section 2.1) could enable higher resolution runs over extended periods in the future.

To assess both the potential future application of MAR v3.9 over the AP as well as the effects of both vertical and horizontal resolution on modelled melt estimates, we compare melt occurrence and flow characteristics from Nov 1, 2004 to March 31, 2005 between multiple versions of the MAR model. This included three versions of v3.9 (Section 2.1), with two 5km and 10km resolution versions run with 24 vertical layers as well as an additional 10km resolution version with 32 vertical layers (10km V). The effect of the enhanced horizontal resolution on topography is substantial; the maximum height of the AP in the 5km version of the model is 2567m, but only 2340m in the 10km version. We find that the effect of increasing horizontal resolution to 5km is to limit the consistent strong melt production just leeward of the AP and that an increase in either horizontal resolution or vertical discretization limits eastward flow

(Fig. S12). As compared to AWS data at the Larsen IS AWS, all MAR configurations largely replicated the dominant southwesterly and northeasterly flow, although we found an enhanced bias for southeasterly flow with the enhanced-resolution versions of the model (Fig. S13). The effects of local topography on wind speed should be relatively limited as the region surrounding the Larsen ice shelf AWS station is relatively flat. Bedmap2 (Fretwell et al., 2013) reports mean (standard deviation) elevation values of 37.38m (0.53m) in the 5km surrounding the station and 37.37m (0.78m) in the 10km surrounding the station. The mean (standard deviation) values for slope are 0.015°(0.018°) at both resolutions. We conclude that a further increase in vertical discretization or horizontal resolution may potentially reduce flow towards the eastern edge of the Larsen C ice shelf, although the effect of better-resolved topography may allow more westerly flow in MAR to cross the AP.

As has been suggested by previous studies (Van Wessem et al., 2015a), the implementation of a non-hydrostatic model may improve the representation of westerly föhn flow over the eastern Larsen Ice Shelf (Hubert Gallée, personal communication). We note that previous work has suggested that a 5km non-hydrostatic model was still unable to capture föhn flow on the eastern portion of the Larsen C ice shelf (according to the AWS records), partially due to the inability to simulate southwesterly föhn jets, and that resolutions as high as 1.5km are required to simulate föhn flow accurately (Turton et al., 2017). However, recent work found that spatial resolutions as high as 2km in the non-hydrostatic WRF model were still unable to fully-resolve the steep surface temperature increases associated with the beginning of föhn flow (Bozkurt et al., 2018), suggesting that neither increased spatial resolution nor a non-hydrostatic model may be sufficient to fully capture the effects of föhn flow. We conclude from the main analysis that reduced eastward propagation of westerly winds may contribute to a lack of MAR melt in the CL region as compared to satellite estimates but that melt just east of the AP (the NL) region is represented with relative accuracy. This is further confirmed by the similarity between the spatial trends for melt occurrence as compared to QuikSCAT estimates. We remind the reader that previous work has suggested that föhn flow occurred only 20% of the time during a single melt season, and that substantial melt occurred in conditions where föhn winds are not present (King et al., 2017), suggesting that other factors contributing to surface melt energy may be equally, if not more, important for developing accurate melt estimates in RCMs. Because the current class of RCMs which employ the hydrostatic assumption, such as MAR, can be run for relatively long periods and contain relatively realistic representations of the snowpack, they can provide additional insights into the cumulative effects of surface melt over multiple seasons, with the understanding that the surface melt produced by föhn flow will likely be under-represented in the eastern regions of the Larsen C ice shelf.

Previous literature has pointed to several limitations in the remote sensing data sources used here which are either intrinsic to the satellite data itself or a product of the algorithm selected for melt detection (Ashcraft and Long, 2006). Products derived from QuikSCAT are limited in temporal resolution because the satellite passes daily, and may therefore ignore sporadic melt occurring at other times of the day. However, previous studies have compared total melt days from the QuikSCAT ft3 algorithm with a measure derived from surface temperature at seven automatic weather stations and shown a positive QuikSCAT ft3 bias compared to AWS (Steiner and Tedesco, 2014). Similarly, all PMW algorithms are limited by a relatively low resolution (25km) and twice-daily passes. Periods of melt occurrence have also been shown to be sensitive to the choice of algorithm (Tedesco 2009).

In future work, we will extend this model run to the 1982-2017 period as well as explore a higher-resolution run
of a newer version of MAR, producing hourly outputs for the near-surface atmosphere. These runs will allow us to
examine the frequency of föhn winds, the concurrent meltwater production and the effects of föhn-induced melt on
the snowpack. We will use this multi-decadal record to examine interannual trends of föhn winds in all seasons as
well as the cumulative effect of a changing regional climate on the snowpack of the NE basin.
Acknowledgements.   This research was funded by the National Science Foundation, grant 1131973. Thanks to the
University of Wisconsin Madison for the use of AWS data, to Dr. Nick Steiner for help with the processing of
QuikSCAT melt occurrence and especially to Dr. Patrick Alexander and Dr. Kate Briggs for their constructive reviews
of early versions of the manuscript.

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

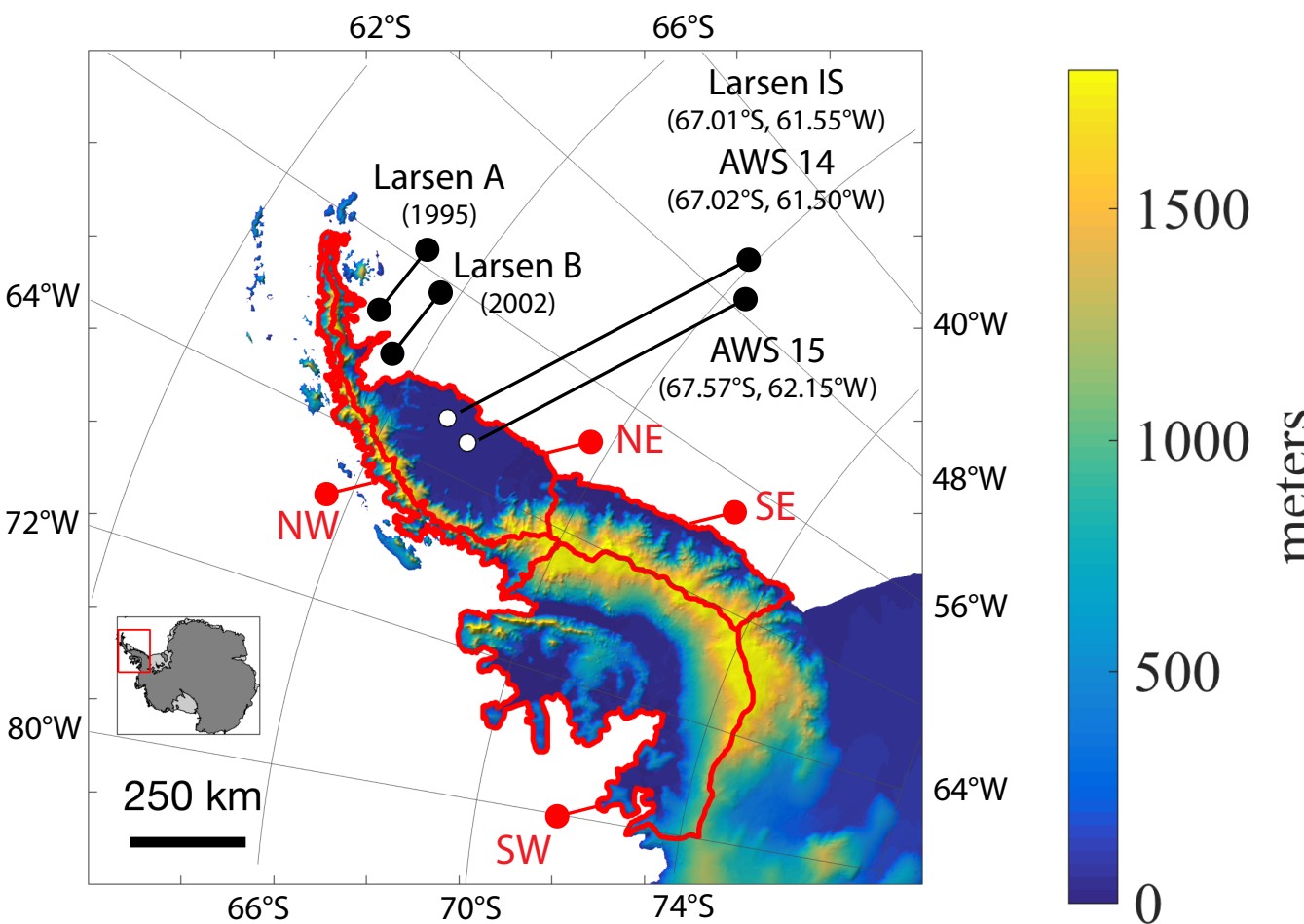

**Figure 1: Full MAR domain showing topographic relief from bedmap2 (https://www.bas.ac.uk/project/bedmap-2) at 1km, former ice shelves with dates of collapse, locations of automatic weather stations (Larsen IS and AWS 14 stations are located within the same MAR gridcell) and basins corresponding to SW (basin 24) NW (basin 25) NE (basin 26), SE (basin 27) from Zwally,et. al. 2012**

| Abbreviation | Definition |
|---|---|
| **MAR model : criteria for melt occurrence (Section 2.1)** | |
| $LWC_{0.4}$ | liquid water content in the first meter is greater than 0.4 mm we (water equivalent) |
| $MF_{0.4}$ | total meltwater production over the day exceeds 0.4 mmwe |
| **Passive microwave : criteria for melt occurrence (Section 2.2.2)** | |
| zwa | threshold based on winter mean temperature brightness, Zwally and Fiegles, 1994 |
| ALA | threshold based on winter mean temperature brightness, Ashcroft and Long, 2006 |
| 240 | fixed threshold method (Tedesco, 2007) |
| PMWAll | Condition when zwa, ALA, 240 all report melt occurrence |
| **Active microwave (QuikSCAT) : criteria for melt occurrence (Section 2.2.1)** | |
| QuikSCAT ft3 | threshold based on winter mean backscatter (Steiner and Tedesco, 2014) |
| **Observation-based regions of high melt occurrence (Section 3.2)** | |
| CL region | high melt at the center-east of the Larsen C ice shelf, melt days exceeding 1 std dev of PMWAll mean melt occurrence |
| NL region | high melt in the north and west of the NE basin, consisting of the NE basin above the mean latitude of CL region which excludes the CL region |
| **Conditions for melt occurrence (Section 4.2)** | |
| PMWEx | PMWAll reports melt occurrence but MAR does not |
| QSEx | QuikSCAT ft3 reports melt occurrence but MAR does not |
| | |
| MAR-R | criteria when MAR data is used only when AWS data is available |
| | |
| | |

1    **Table 1: Abbreviations used throughout text**

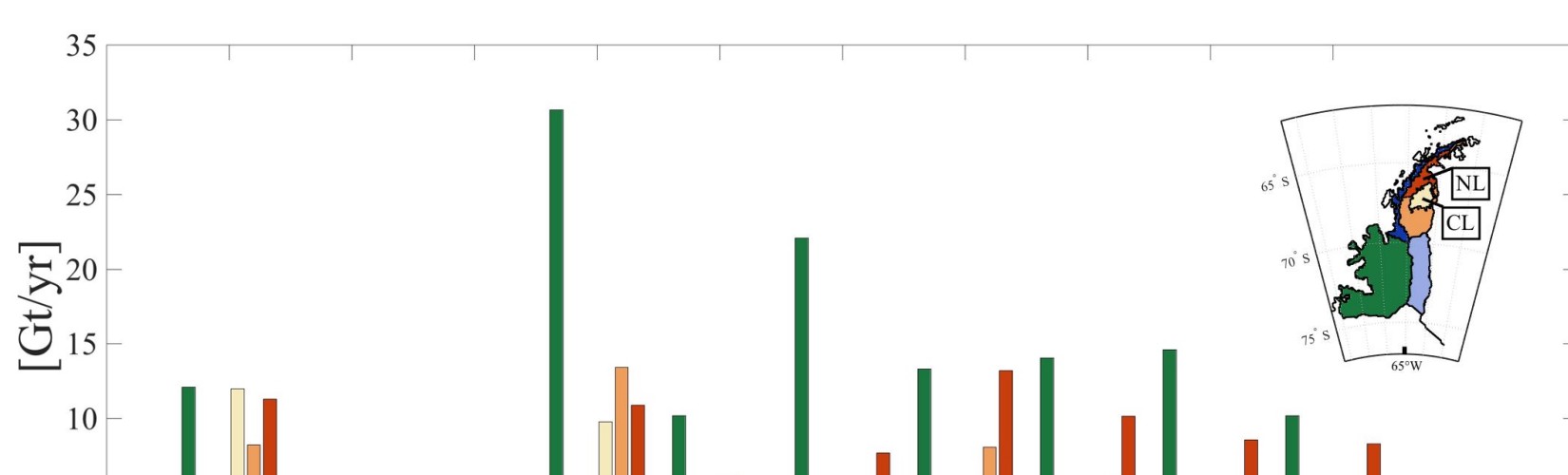

**Figure 2 Annual meltwater production from MAR [Gt/yr] shown for masks shown in inset ('2001' corresponds to meltwater production from July 2000- June 2001. NW, SW, SE basins are shown as in Fig. 1. NE basin is divided into the NL mask, the CL mask and the remaining portion of the NE basin (NE – (CL+NL)). The CL and NL masks are described in text.**

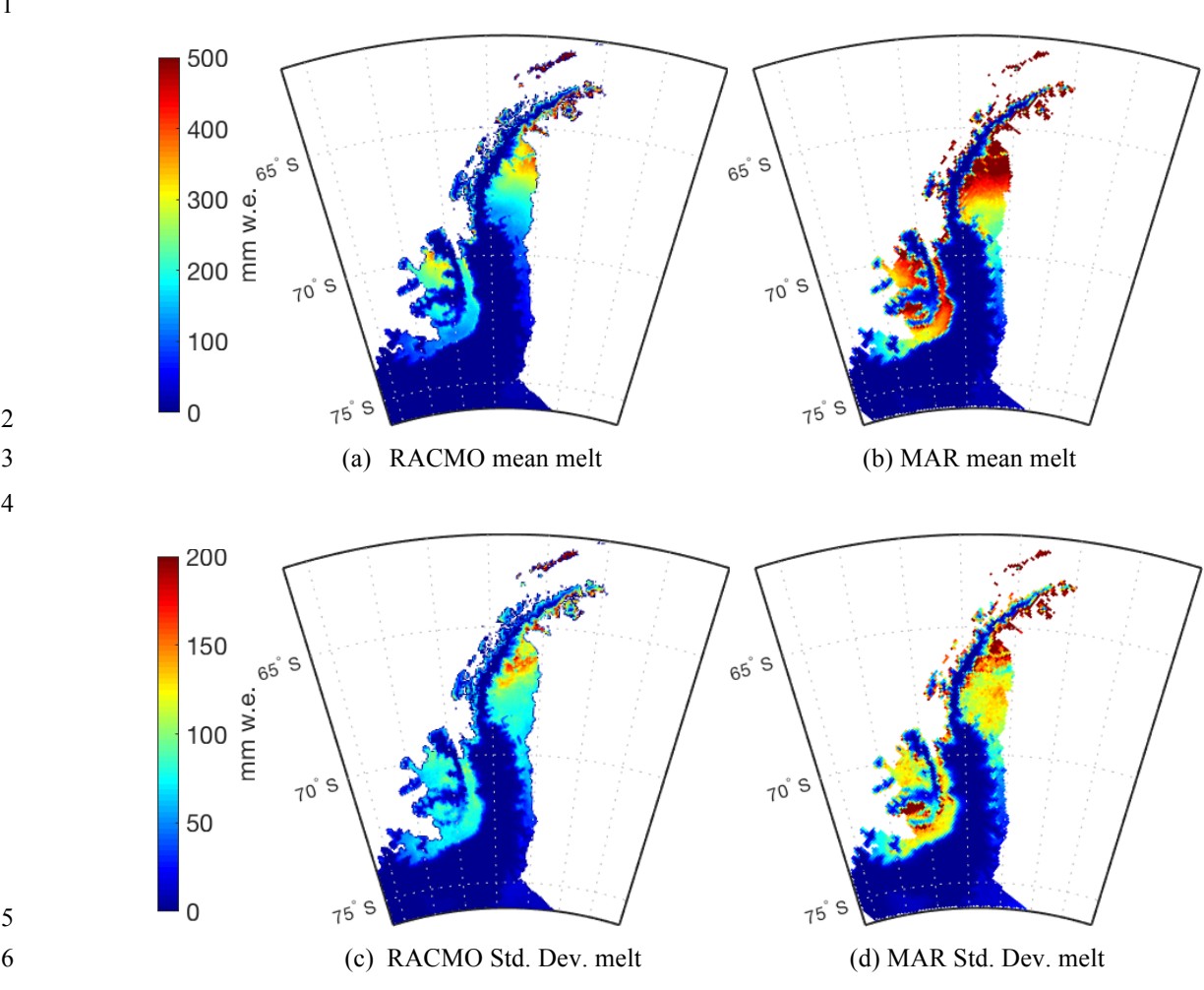

(a)  RACMO mean melt          (b) MAR mean melt

(c)  RACMO Std. Dev. melt          (d) MAR Std. Dev. melt

**Figure 3 Meltwater production (2000-2009). RACMO2.3p2 at 5.5 km resolution, mean annual meltwater production (a) and standard deviation (c) and MAR v. 3.5.2 at a 10km resolution, mean annual meltwater production (b) and standard deviation (d)**

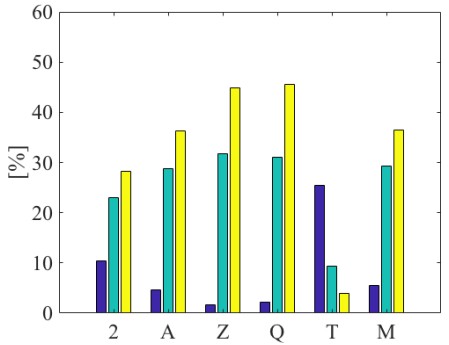

(a) Melt Occurrence at the Larsen Ice Shelf AWS Threshold = 0.4 mm w.e.

**MAXT2m(AWS) < 0°C**

**MAXT2m(AWS) ≥ 0°C**

AvgT2m
(MAR-AWS)

(b) AvgT2m bias

(c) AvgT2m bias

**Data Sources:**

Satellite
2    PMW 240
A    PMW ALA
Z    PMW zwa
Q    QuikSCAT

AWS-based
T    AvgT2m > 0°C
M    MaxT2m > 0°C

**Melt Occurrence:**

    MAR Only
    MAR & Obs
    Obs. Only

MaxT2m
(MAR-AWS)

(d) MaxT2m bias

(e) MaxT2m bias

**Figure 4 Melt Occurrence and Temperature Biases at the Larsen Ice Shelf AWS Station. Percentage of total days (DJF, 2001-2010) showing melt occurrence from observational sources as compared to MAR v3.5.3 melt occurrence using the $MF_{0.4}$ metric (a) Temperature biases (MAR-AWS) for AvgT2m (b,c) and MaxT2m (d,e) when Max T2m is less than 0°C (b,d) or greater than 0°C (c,e)**

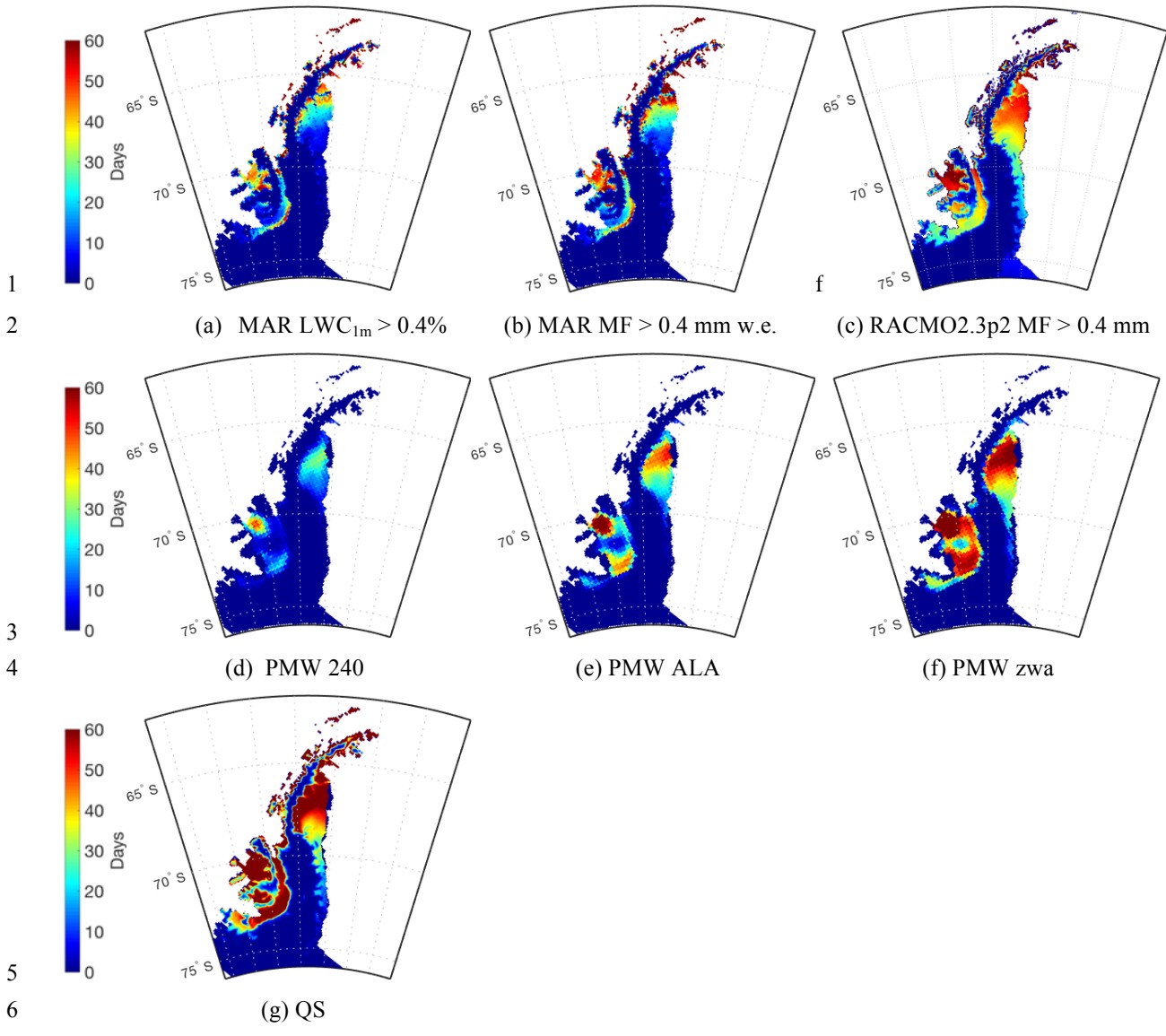

2              (a)   MAR LWC$_{1m}$ > 0.4%         (b) MAR MF > 0.4 mm w.e.        (c) RACMO2.3p2 MF > 0.4 mm

4                  (d)  PMW 240                  (e) PMW ALA                  (f) PMW zwa

6                  (g) QS

**Figure 5 Average number of melt days (2000-2009) from multiple sources (a) MAR, Liquid Water Content > 0.4% for three**
**consecutive days. (b) MAR Total Melt Flux >0. 4 mm w.e. for 1 day or more (c) RACMO2.3p2, Melt Flux > 0.4 mm w.e.**
**Satellite-based metrics include (d) PMW 240 algorithm (e) PMW ALA (f) PMW Zwa (g) QuikSCAT. All satellite-based**
**estimates include a melt day only when part of a sustained three-day period of melt.**

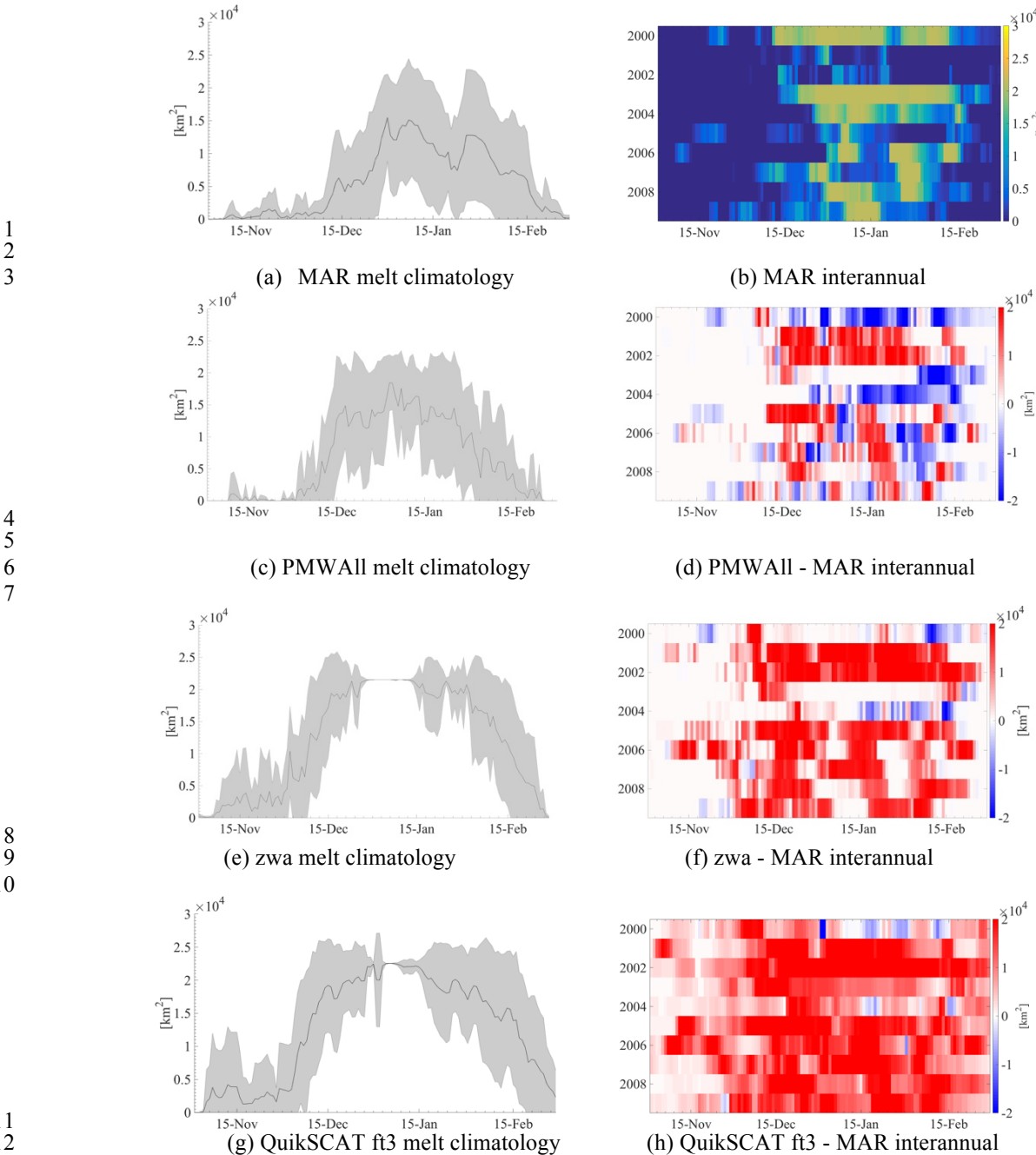

**Figure 6 CL-region, described in text and shown in inset in for (a), average and inter-annual melt occurrence in MAR, PMW and QuikSCAT data. (a) MF0.4 melt extent climatology with one standard deviation shown in grey envelope (b) melt extent for MF0.4 from 1999-2009 (c) melt climatology PMW All (d) interannual difference melt extent PMWAll - MAR (e) melt climatology PMW zwa (f) interannual difference in melt extent PMWzwal – MAR (g) melt climatology QuikSCAT ft3 (h) interannual difference in melt extent QuikSCAT ft3 - MAR**

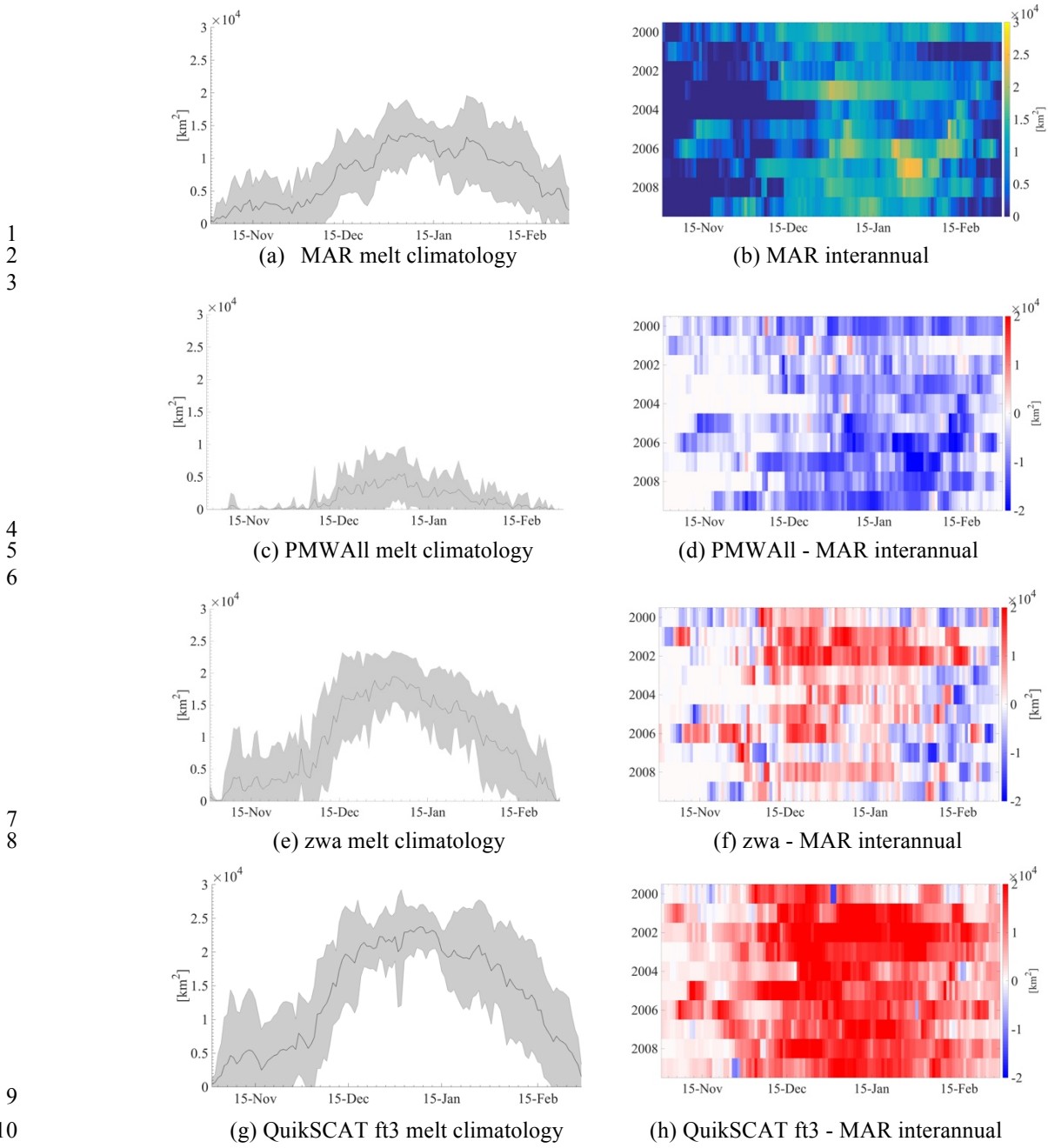

**Figure 7 NL-region, described in text and shown inset in (c), average and inter-annual melt occurrence in MAR, PMW and QuikSCAT data. (a) $MF_{0.4}$ melt extent climatology with one standard deviation shown in grey envelope (b) melt extent for $MF_{0.4}$ from 1999-2009 (c) melt climatology PMW All (d) interannual difference melt extent PMWAll - MAR (e) melt climatology PMW zwa (f) interannual difference in melt extent PMWzwal – MAR (g) melt climatology QuikSCAT ft3 (h) interannual difference in melt extent QuikSCAT ft3 - MAR**

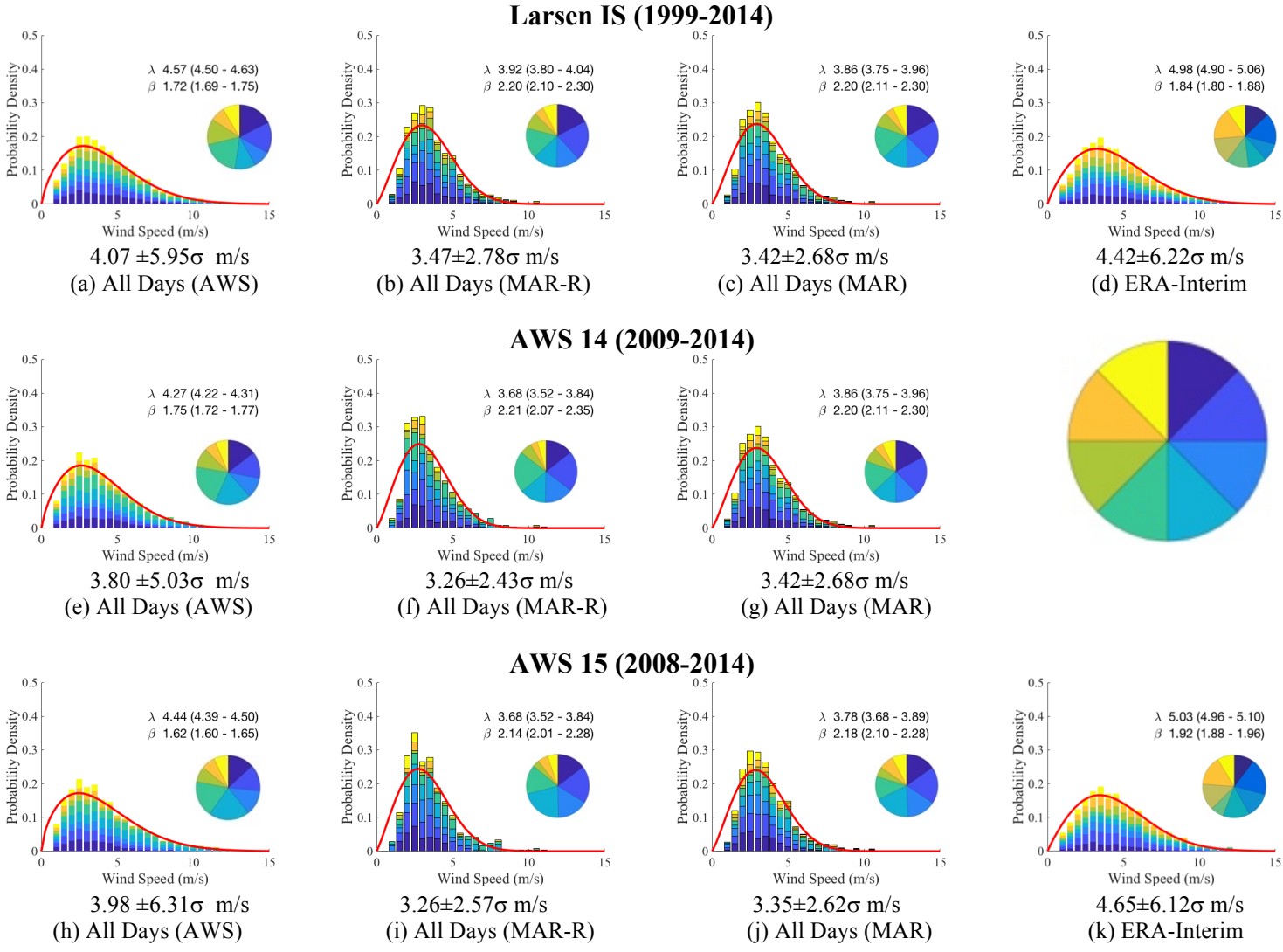

**Figure 8 Probability distribution (y-axis) of summer (DJF) wind speeds (x-axis) and direction proportions inset. Wind directions corresponding to colors in 45° increments shown right of (g). Curve shows Weibull curve shape (β) and scale (λ ,m/s). Datasets for AWS (col 1), MAR-R (col 2), MAR from 1999-2014 period (col 3) and ERA-Interim for the AWS-restricted period (col 4). Shown for station Larsen IS ( row1, a,b,c,d), AWS 14 (row 2, e,f,g), AWS 15 (row 3, h,i,j,k) Values below figures are expected values.**

| | NE (0°-90°) | SE (90°-180°) | SW (180°-270°) | NW (270°-360°) |
|---|---|---|---|---|
| **DJF All Days** | | | | |
| **MAR shows wind direction** | | | | |
| MAR percentage | 39.0% | 23.5% | 23.2% | 14.1% |
| MAR expected wind speed [m/s] | 3.48(±2.46) | 3.47(±2.62) | 4.46(±4.44) | 3.66(±4.69) |
| AWS expected wind speed [m/s] | 3.79(±4.35) | 4.19(±6.01) | 5.35(±9.16) | 4.00(±7.63) |
| **AWS shows wind direction** | | | | |
| AWS percentage | 34.3% | 17.4% | 28.9% | 19.3% |
| MAR expected wind speed [m/s] | 3.47(±2.49) | 3.49(±2.14) | 3.86(±3.54) | 6.40(±10.14) |
| AWS expected wind speed [m/s] | 3.96(±4.65) | 3.77(±4.97) | 4.77(±7.89) | 6.70(±16.94) |
| **Temp. biases (MAR-AWS)** | | | | |
| Avg T2m | 0.68°C | 0.65°C | 0.94°C | 0.72°C |
| Max T2m | -2.16°C | -1.40°C | -1.19°C | -2.35°C |
| **Temp. bias where T2m > 0°C (MAR-AWS)** | | | | |
| Avg T2m | -1.36°C | -1.50°C | -1.06°C | -1.06°C |
| Max T2m | -2.96°C | -3.05°C | -2.33°C | -2.75°C |
| | | | | |
| **DJF, MAR reports melt** | | | | |
| MAR wind direction percentage | 34.7% | 27.6% | 24.5% | 13.2% |
| AWS wind direction percentage | 35.2% | 13.9% | 25.6% | 25.2% |
| **Temp. biases (MAR-AWS)** | | | | |
| Avg T2m | 0.77°C | 0.56°C | 1.05°C | 0.52°C |
| Max T2m | -2.11°C | -2.20°C | -0.95°C | -1.43°C |
| **Temp. bias where T2m > 0°C (MAR-AWS)** | | | | |
| Avg T2m | -0.93°C | -1.13°C | -0.53°C | -0.98°C |
| Max T2m | -2.57°C | -3.16°C | -1.66C | -1.61°C |

**Table 2: Proportions for wind direction and associated temperature biases at the Larsen Ice Shelf AWS station from 2000-2009 restricted to the summer season (DJF)**

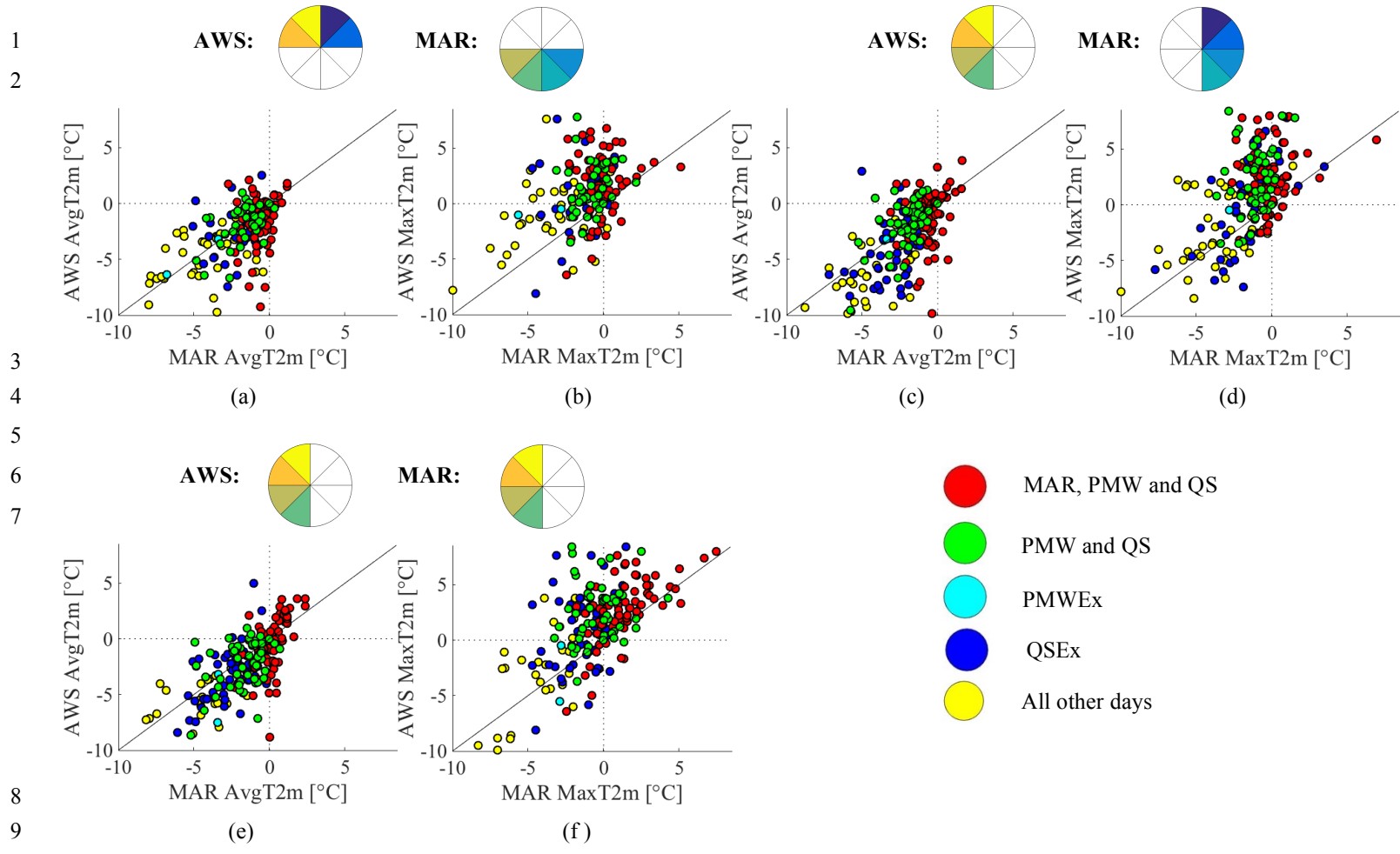

**Figure 9 MAR vs AWS temperatures at the Larsen Ice Shelf AWS station for DJF from 2001-2009 for melt occurrence criteria as shown bottom-right and described in text. Wind direction biases are shown for when northerly AWS flow is reported as southerly in MAR (a) AvgT2m (b) MaxT2m, when westerly AWS flow is reported as easterly in MAR (c) AvgT2m (d) MaxT2 and when AWS and MAR both report westerly flow (e) AvgT2m (f) MaxT2m.**

2
