# Peer review of "Melting over the Northeast Antarctic Peninsula (1999-2009): evaluation of a high-resolution regional climate model"

_The Cryosphere, 2017_

## Referee Comment (RC1) · Anonymous Referee #1 · 18 Jan 2018

Summary:

This paper focuses on melting over the East Antarctic Peninsula, with specific focus on the Larsen C ice shelf using the Modele Atmospherique Regionale (MAR) model. Model results are evaluated by satellite estimates of melt from passive and active microwave sensors, and from automatic weather station observations of near-surface variables. The abstract and introduction point to an assessment of the impact of westerly föhn flow on the melting, however this doesn't seem to have the same prominence in the results and discussion. My main concern is that the focus of the paper is to evaluate the MAR model in terms of melting, however there appears to be relatively

little discussion on the impact of the chosen model options, such as the horizontal and vertical resolution, which limits the readers understanding of how efficient MAR is at reproducing melt on the ice shelf, as seen in satellite observations. The potential impact of model choice, and model physics is most clear in the results and discussion of the wind direction, where there is a large discrepancy between the model and observations. I think this work is of sufficient merit and in terms of comparison of the model with both AWS and satellite observations, it takes a novel approach. Similarly, this (according to the authors) is the first time that MAR has been applied and evaluated over the Antarctic Peninsula, as opposed to Greenland, and therefore the results of this paper could influence model development by testing it in complex regions. As a model evaluation paper, it uses multiple data sources over a relatively long-time period. However, the model resolution is rather coarse over this region, as higher resolution model studies have been performed previously. An additional novel aspect which this paper does not mention is the length of this modelling study. Output for the model are for 15 years, which is a long study period over this region. Previous melt-föhn studies have largely focused on case studies, or shorter time periods (e.g Elvidge et al 2015, Grosvenor et al (2014), King et al 2017). More emphasis could be put on the length of study, as this is of importance. In my opinion, this work is sufficient for publication, but major revisions are required beforehand.

Major concerns:

The major concerns are mostly centred around the choice of model, and the justification for this model. There are a number of possible reasons why this model (in its current state) should not be used for this area (coarse resolution, hydrostatic assumption, low vertical resolution). However, if the goal of this study is to evaluate MAR to assess whether it is useful to model melt in this area (and particularly the wind-driven melt), then it needs to be phrased differently in the abstract and introduction. If an objective is to test whether hydrostatic models can capture the wind-induced melting and wind processes (as opposed to non-hydrostatic which have been largely used), then

this needs to be outlined. Similarly, if an objective is to assess whether you need to go to much lower resolutions, or whether you can model melt at coarser resolutions, this should be stated. Currently, the paper reads that you have reservations about the potential of the model before you start, but continue to use it anyway. In the discussion, you would also need to compare this study to ones of higher resolution or different models, to say whether MAR has been successful at capturing wind-induced melt. The following points, break down the concerns with the model set up.

The hydrostatic assumption and horizontal resolution of MAR. In the abstract, authors state that "melting in the East AP can be initiated by both sporadic westerly föhn flow over the AP and by northerly winds advecting warm air from lower latitudes. To assess MAR's ability to simulate these physical processes, this study..." (line Pg 1, 24-27). Then later in the discussion you state that MAR can't accurately represent the wind direction and föhn processes due to the model's hydrostatic assumption, and state that a non-hydrostatic model would do better (pg 15 line 23-27). Models which have previously, successfully captured the föhn characteristics (WRF and UM) are non-hydrostatic, and this appears to be known to the authors prior to the study as in the introduction (Pg3, line 33), they discuss the (non-hydrostatic) RACMO study, and justify their use of a coarser resolution due to the hydrostatic assumption. If a large part of the study is to assess the impact of wind on the melting, why chose a model which can't represent the dominant westerly flow (and subsequent downward föhn flow) over the AP? If an objective of this study was to attempt to model this type of flow using a model with hydrostatic assumption, then this should be made clearer, and authors should note any previous studies of this kind.

The above comment is also linked to the relatively coarse (for this region and this topic) horizontal resolution used here. Previous föhn studies use much finer resolution (5km, 1.5km) and suggest that this resolution is required for adequate föhn representation. Authors discuss the Van Wessem et al (2015) study which suggests higher resolution than 5.5km. However the authors appear to use the following statement: "where hydrostatic assumption is preserved (such as this model run), higher resolutions may inhibit flow in the model..." (pg 3, line 35/36) to justify using a lower horizontal resolution. To address this issue, I think a sensitivity study using higher resolution is required. It doesn't have to be for the full-time period, but should capture at least a season of melt to assess whether the spatial resolution could improve the results, and whether the breakdown of the hydrostatic equation does limit air flow. You should also add more to the discussion about this. The spatial resolution is not mentioned at all in the discussion, and as found in other papers (Van Wessem et al 2015, Turton et al 2017, Elvidge et al 2015), higher resolution runs do capture föhn winds.

In the abstract and introduction, a fair amount of emphasis is put on the role of föhn winds and northwesterly winds e.g (Pg 1, line 25, 32, 33, 35, Pg 3, line 5-21, 34-37, Pg 4, line 1, 20). However, in the results and discussion, this is not discussed thoroughly, either in the context of other studies, or how well MAR can model these features. More discussion of the föhn characteristics and melt related effects needs to be included in your discussion to have such a prevalence in the earlier sections.

The abstract states that increased spatial resolution and topographic resolution could improve the output from MAR, but there is no mention of this in the discussion or results. You should not include statements in the abstract which do not reflect the results of the study. Either address whether changing the spatial or topographical resolution does impact the modelled melt or near-surface conditions, or remove this from the abstract.

The vertical resolution of MAR's atmosphere is very coarse, especially to have the lowest model level at 2m above the surface (pg 5, lines 10 and pg7 line 29/30). What is the vertical discretisation of your levels? The WRF model for instance has difficulties if there is over 1km between model levels, or if the stretching factor is greater than 20%. Similar to the first major comment, a sensitivity study is required to assess the impact of this vertical resolution on the representation of the near-surface conditions and the wind. This is much coarser than many studies of this kind and studies using

[Figure]

MAR (see for example, Gallee et al 2015 or Wyard et al 2016 who both use 60 vertical levels). Again, this doesn't need to be the full period (and shouldn't be as this would be a huge/long undertaking) but a full season should be tested using a number of higher vertical levels.

Minor comments:

Lots of sentences start with 'Because'... Perhaps vary this a little.

Coordinates are needed on Figure 1 and 2.

The insert for Figures 3 and 4 is very small and hard to read. Could it maybe be included in Figure 1 alongside the other map? Or move Figure 8 earlier, where the insert is clearer.

There needs to be a discussion of why MAR is unable to capture the wind direction, as presented in Section 4.2. Is it getting the synoptic situation wrong? Or near-surface conditions? Is this related to the relatively coarse resolution? This section is important for assessing the impact of the wind direction in MAR, but it should also be stated that if MAR is getting something like large scale flow wrong, it might be getting other processes wrong due to this.

Detailed comments:

Abstract:

Line 34: Authors state that reducing the underestimation of flow may be obtained by increasing the spatial resolution, but this is not given much discussion later in the paper. Either remove it and focus on hydrostatic assumption, or include changes to the spatial resolution in the discussion- either results from the suggested sensitivity study, or by discussing other studies.

Line 35: You mention reducing the underestimation of flow may be obtained by using higher-resolution topography, but this is not mentioned anywhere else in the paper.

Similarly, you do not state what topography is used in the model, or what resolution it is.

Introduction:

Pg 2 Line 7: remove 'finally' .

Pg 2 Line 23: 'suggested' should be 'suggest' .

Pg 3 Line 17: Remove 'during recent warming' at the end of the sentence.

Pg 3 Line 20: Is there a citation for this? 'East AP is as vulnerable to wind dynamics as it is to temperature change'. Has a study quantified the difference in vulnerability? What vulnerability mean in this context?

Pg 3/4 : Some citations are missing which may need including here, such as King et al 2017 and Elvidge et al 2015 which discuss föhn and melting on Larsen C.

Pg 4 Line 2/3: 'These last studies taken together' doesn't read well. Perhaps change to 'Both of these studies, along with others by Elvidge et al 2015 and King et al 2017, discuss both the atmospheric. . .'. This would include the previous comment also.

Pg 4 Line 10: AWS is not defined yet (but is later defined on line 13/14).

Pg 4 Line 14: Which satellites? Just give their names/abbreviations here.

Pg 4 Line 15 and 20: Be consistent with use of abbreviations or names. AP for example.

Pg 4: Line 24: from what date to 2014?

Data and Methods:

Pg 4 Line 27/28: MAR and AWS have been defined earlier.

Pg 5 Line 5: Which part of Antarctica?

Pg 5, section 2.1: Where can readers can get more information about MAR, such as physics set up? Include a citation for this. What is the model top? 23 Sigma layers

is very coarse (see major comments). Why was this vertical resolution used? Only 1 domain or is it nested? What is the resolution of the topography, and what dataset is used? BEDMAP2 for instance?

Pg 5 Line 17: what mask? Land use? Land/sea?

Pg 5 Line 20-26: reorder this paragraph to make it clearer what each notation is. For example, line 20-23, both notations are stated, but more emphasis is put on LWC0.4. It could be split into 2 sentences, one for LWC0.4 and one for MF0.4.

Pg 5 Line 25: What is the justification for this condition? The same as LWC0.4 (Tedesco et al 2007)?

Pg 5 Line 30: change 'microwave sensors are weakly affected. . .' to 'microwave sensors are only weakly affected'.

Pg 5 line 31: after citation, change 'where' to 'whereas'.

Pg 6, line 6/7: 'used extensively' is stated, but there is only 1 citation. Are there other important citations? The Drinkwater and Liu (2000) reference only looks at Antarctica, not Greenland.

Pg 6, equation 1: what is Tc?

Pg 6: active and passive microwave: what are the spatial resolutions of the satellites? To allow some comparison with the 10km resolution of MAR.

Pg 7, line 2: confused what 'here' means in this context. For this location?

Pg 7, line 3: 'zwa is based on the winter mean threshold'. Threshold of what? Winter mean air temperature?

Pg 7, line 8: pressure observations are not mentioned here but they are in line 27 onwards.

Pg 7, line 19: what is meant by 'expected'? This is also used in terms of wind speed

later in the paper, and I don't understand its use.

Pg 7, line 27: What is meant by 'estimating' pressure from the AWS? Is pressure observed by the AWS or not? Pressure is also not mentioned elsewhere in the paper, so if it is not used, remove it.

Pg 7, line 28: remove 'a' from 'also estimated at a approximately...'

Pg 7, line 29: your lowest model level is 2m but only 23 sigma levels are used- this is very coarse. See major comments above. Are 2m diagnostics output from MAR? As you could use these instead of taking it from the lowest model level, if this should change when you run the sensitivity study for varying the number of vertical levels.

Results:

Pg 8, line 1: 'assess the extent to which each station is representative of larger scale climate variability'. Even though AWS14/Larsen and AWS15 are so close together? Do they have a different extent?

Pg 8, line 13: keep consistent with abbreviations.

Pg 9, line 3/4: you state coordinates/latitudes in the text but there are no coordinates on your Figure 2 plots. Include coordinates on the plots.

Pg 9, line 19: 'data sources ad secondarily' should read 'data sources and secondarily'.

Pg 9, line 19: What are the spatial resolutions of the data sources? You mention this, but then don't go into it any further. However, you mention the depths presumed for melt water content and then discuss it for the next paragraph. Perhaps more information on the spatial resolutions is needed.

Pg 9, line 28: Give some examples of these 'low' melt occurrence regions. From elevation information in supplement table 1, they aren't on the ice shelf, are they on the main spine of the AP?

Pg 9, line 28: heterogeneous in what way? Elevation? Surface type?

Pg 9, line 37: what is 'N column'?

Pg 10, line 17: what is PMWAll-coincident?

Pg 10, line 31: 'early pulse around Dec 15th', do you mean Nov 15th? As there are small pulses of melt here, and December 15th melt looks much larger.

Pg 11, line 25: 'during that period'. Which period? Be more specific.

Pg 12, line 4: remove 'station' after AWS.

Pg 12, line 8: remind readers of MAR-R/MAR differences here.

Pg 12, line 15/16: 'demonstrate the consistency of wind biases' and 'how wind biases vary by latitude' are slightly contradictory. Are they consistent or variable?

Pg 12, line 16: remove 'whereas', as you aren't comparing AWS and MAR, as one is for low wind and the other for high wind speeds.

Pg 12, line 16: 'MAR is dominated by northerly winds'.

Pg 12, line 27/28: Might be useful to highlight which rows of the table you mean here. When comparing all times and melt times. It isn't immediately clear that 'increased N and W flows' means compared to when there all days are included.

Pg 12, line 34: citation style.

Pg 12, line 36: the abbreviation Ts is used in the table for when temperature is >0degC. However, in the text you say that when 2m-temperatures exceed 0degC. Stick to the T2m abbreviation.

Pg 13, line 5: remove extra space before -3.04.

Pg 13, line 8-12: include reference to figures here.

Pg 13, section 'observed NE flow and observed SW flow': It needs to be clearer that
when MAR has different wind directions to the observations, MAR is wrong. Especially in the case where there are large differences (NE vs NW for instance). And explain what the possible reasons are for this. Is MAR not getting the synoptic scale wind direction right? Or is there not enough blocking on the west of the AP to prevent flow over the AP when there shouldn't be? This section is a good idea to see what impact the wind direction is having in MAR, but it should also be stated that if MAR is getting something like large scale flow wrong, it might be getting other processes wrong due to this.

Pg 14, line 1-6: In this section, it might be good to remind the reader, that in case 2, MAR is getting the wind direction wrong when compared to AWS. So that the reader can put these results into context.

Pg 14, line 7: Using Ts abbreviation but you have only talked about air temperature and used T2m previously.

Pg 14, line 11-13: confusing sentence. What is meant by expected?

Pg 14, line 13: I don't think figure 6 e-h are necessary. They are not discussed as much in the text, and the information is given by the 6a-d. Similarly, figure 7 could be included into figure 6 in place of 6e-6h.

Discussion:

Pg 14, line 30: remove 'in the aggregate'.

Pg 15, line 4: where should be when.

Pg 15, line 6/7: Any suggestions for why there are less westerly winds in MAR?

Pg 15, line 17-21: considering the impact of föhn winds is prominent in the abstract and introduction, this seems like a short discussion of them. See major comments.

Pg 15, line 19-21: wind speed may not be the biggest issue here if MAR is unable to get wind direction right.

Pg 15, line 23-25: include references to and discussion of non-hydrostatic models that have captured föhn flow- e.g Elvidge et al, 2015 (Met UM model), Turton et al 2017 (WRF model).

Pg 15, general: The abstract suggests that increasing the spatial resolution of MAR or the topography in the model may improve output, but this isn't discussed in your discussion. See major comment.

Pg 16, line 7: Figure 8 should come earlier in the text. This is a good summary figure and could be included in page 11 where interannual variability is mentioned.

Pg 16, line 19/20: 'melt in the NL region is particularly sensitive to föhn induced melt'. You need to support this with other studies (e.g Elvidge, et al 2015, Cape et al 2015), as your study only mentions föhn jets on the SW of the ice shelf in earlier discussion.

Pg 16, line 22/23: is this future work? As this study doesn't talk about large-scale atmospheric drivers at all. Or you need to support this with studies which look at large-scale atmospheric patterns and their related wind patterns in this region (such as Cape et al 2015).

Figures:

Figure 1: include in the caption that Larsen IS and AWS14 have the same MAR grid cell, which is why they are on the same marker.

Figure 1: Where is the topography data from?

Figure 1: include coordinates.

Figure 2: include coordinates.

Figure 3/4: make insert bigger, or include it in figure 1.

Figure 6: make a heading over a/b 'Case 1' and over c/d 'Case 2'. I don't think anything else is gained from e-h, as they are mentioned only briefly in the text.

Figure 6: g and h are not described in the caption.

Figure 6/7: 'yellow as only shown for g,h'. Not sure what this means, as yellow markers are used in every subplot, not just g and h.

Figure 7: could be combined with Figure 6.

Figure 8: if this goes earlier in the text, then the size of the insert is sufficient for the other figures which require it.

Supplementary Figure 6: lettering is not right. There is no a-c as in the figure, and g-m are not in the caption. There are only 6 subplots, so I assume a-f is correct.

Table:

Table 1: Ts should be T2m, unless actual surface temperature data is being used, but is not mentioned elsewhere in the paper.

Typos:

Pg 1, Line 29: satellites should be satellite

Pg 2, line 21: comma after citation

Pg 2, line 27: comma after citation

Pg 4, line 20: umlaut missing over o in föhn

Pg 5, line 31: comma after citation

Pg 6, line 6: remove full stop after algorithm, there is one after the citation.

---

## Referee Comment (RC2) · Anonymous Referee #2 · 5 Apr 2018

General:

This study presents an evaluation of the regional climate model Modèle Atmosphérique Régionale (MAR) in simulating surface melting over the northern East Antarctic Peninsula for the period 1999-2009. MAR has been used for the first time over that region at a spatial resolution of 10km. In addition, near-surface air temperature and wind speed/direction is taken into account in order to assess the model performance of these important drivers in surface melting. As observational reference data satellite estimates from passive and active microwave data and three automatic weather stations is used.

[Figure]

Major Comments:

The manuscript provides an interesting and relevant topic in regional climate model evaluation of high resolution climate simulations for snowpack and its melting over the northern East Antarctic Peninsula. The authors present a good overview of used datasets, experimental setup and an interesting validation of melt extent and duration using satellite estimates of active and passive microwave data in combination with station-based near-surface air temperature and wind data to assess underlying processes.

However, in my point of view it would be good to consider the influence of driving reanalysis on the regional climate model results. In addition, the results should be discussed in more detail in the context of other model studies. It would revalue the paper if clear research questions are stated at the beginning and answered at the end.

What about observational uncertainty of satellite data or uncertainties introduced by the postprocessing of satellite data? Would it be possible to include a specific error-estimate to better evaluate the model results and to take into account the observational uncertainty?

What about the impact of ERA-Interim as driving reanalysis data? Would it be possible to add it in the evaluation? Could the mentioned aspects of wind biases and thus resulting biases of melt occurrence have also their origin in the obtained large-scale atmospheric information given by the boundary condition?

Could the mentioned cold bias in MAR (when maximum temperature and average daily temperature exceed 0 degree Celsius) origin from other model deficiencies as well? So far only wind is considered.

In Section 4.2 and 4.2.1 there are many abbreviations introduced which makes it a bit difficult to read. Would it be possible to already introduce those in the methods part and provide a table as overview? Or maybe it is possible to reduce the amount of

abbreviations used in the text.

It would be interesting to discuss presented results (e.g. the underestimation in melting in the center and east of the Larsen C ice shelf) in greater detail to other studies e.g. to other regional climate model studies over the Antarctic region or in general in terms of e.g. issues in snow melting (e.g. onset and ending) in other regions. Also GCMs might have similar issues that would be of interest to consider.

Minor Comments:

Page 3 l. 29 + l. 33: use same space before unit

Page 4 l. 20: change to föhn

Page 5 in section 2.1: Please mention the size of the model domain

Page 5 l. 2: explain abbreviation RCM

Page 7 l. 6: add space after where

Page 7 l.35: remove space before Wilks

Page 12 l. 34: citation with 2 brackets

Page 34 l. 34: remove second brackets

Page 18 l. 8: remove slash in Royal

Page 24 l. 5: add space before Greenland

Figures:

Fig. 1: Please add coordinates to the axes

Fig. 2: Please add coordinates to the axes

Fig: 3: Please have a consistent labeling of axes throughout all the figures 1-8; variable [unit]

Fig: 4: Same as Fig. 4

Fig. 7: ended with a comma

Recommendations:

As some aspects need to be considered, I think this manuscript is not yet ready for publication. Therefore, I would recommend 'major revisions' adding some more aspects with respect to the driving reanalysis and in the discussion to put the presented results in a broader context. A future version with a more concise introduction stating clearly the research questions, the consideration of influence of the boundary data on the results and an improved discussion section could make an interesting contribution.

---

## Author Comment (AC2) · 14 Jun 2018

**Major Changes**

There are 8 major revisions below. in addition to a change to the title to make it more specific to the region.

Revisions #1-5 are in direct response to reviewers' comments.

Revision #6 (regarding wind direction) is in response to a bug that we discovered in the process of revisions.

Revisions #7, #8 are additional changes which followed partially as a consequence of comments from Reviewer #2, although they were not explicitly requested).

In addition, changes in the language for clarity throughout (especially the abstract) and several additional references have been included in light of developments in the subject area in the last few months (King et al., 2017; Weisenekker et al. 2018; Van Wessem et al., 2018; Bozkurt et al., 2018).

**MAJOR REVISIONS**

**1) Discussion of the choice of horizontal and vertical resolution**

We now include a comparison between higher-resolution runs of a newer version MAR over the 2004-2005 melt season. The comparison includes 3 versions of MAR: (a) at 10km (b) where the horizontal resolution is increased from 10km to 5km and (c) where the vertical discretization is increased from 23 to 32 sigma layers. The variables examined are meltwater production, melt occurrence (over the domain) and wind speed/direction at the Larsen Ice Shelf AWS location. We find that an increase in resolution limits melt over the Larsen C ice shelf and increases southeasterly flow, suggesting that while the hydrostatic assumption is kept, the effect of increased resolution will lead to reduced melt overall, but potentially enhance the accuracy of melt just east of the AP due to better-resolved topography. This is specifically presented in the Abstract, discussed in detail in the Introduction and presented in the Discussion and Conclusions.

Abstract: P1, L29-31 Introduction: P3, L 4-12, 26-36 Data and Methods: P5, L 30-37 Discussion and Conclusion: P17, L30 – P18, L9 Supplement Fig. S12, S13

**2) The hydrostatic assumption**

We have altered the text to emphasize the relative advantages/disadvantages of hydrostatic vs. non-hydrostatic versions of the model, i.e. the accuracy of winds (in non-hydrostatic models such as WRF) vs. the long run periods and sophistication of the snowpack (in hydrostatic models such as MAR or RACMO2.3p2). Discussion of the latter is provided in #5 (below)

Re: non-hydrostatic models

We now include a more thorough review of recent non-hydrostatic modeling studies (King et al., 2017; Turton et al., 2018; Bozkurt et al., 2018) noting that factors other than föhn melt are important over the Larsen C ice shelf as well as recent work showing that even a high-resolution non-hydrostatic model was not fully able to resolve föhn characteristics.

Introduction: P3, L 2-12 Discussion and Conclusion: P18, L10-29

**3) Overemphasis on föhn winds**

The original discussion about föhn winds has been substantially limited to where the main emphasis is placed on previous studies with a non-hydrostatic model. More emphasis is placed here on the distinction between the initial intrusion of föhn flow (which high-resolution hydrostatic models may capture) vs. the eastward propagation towards the edge of the Larsen C ice shelf (where the comparison with AWS stations are conducted).

However, a section on northwesterly flow biases is specifically included to address the effects of probably föhn flow

Introduction: P3, L 1-12 Results: P16, L7-20 Discussion and Conclusions: P18, L 18-20

**4) Driving Reanalysis**

To understand the impact of forcing on the representation of wind dynamics in MAR, we have included a comparison of ERA-Interim wind fields and discussed possible reasons for the differences.

Fig. 8 d,k Data and Methods: P5, L 26-28 Results: P14, L35 – P15 7

**5) Comparisons with the hydrostatic model RACMO2.3p2**

We have corrected a typo mis-stating that RACMO2.3p2 is a non-hydrostatic model, added greater detail about recent publications with RACMO2.3p2 over the AP, and included a direct comparison of melt occurrence/meltwater production between RACMO and MAR. References to recent work on RACMO3.2p2 over the AP are included (Van Wessem et al., 2018; Weisenekker et al., 2018) Introduction: P3, 26-36, P4, 7-9 Data and Methods: P 6, 19-23 Results: P10, L 3-19; P11, L18-21 Fig. 3, Fig. 5c

**6) Computation of Wind Direction**

In the process of addressing the major revisions, we discovered a bug with the computation of wind direction in MAR. The now-corrected computation of wind direction substantially reduced the wind direction biases, although we address the biases that are present in light of how the absence of westerly flow affects melt on the eastern Larsen C ice shelf (where the AWSs are located).

Fig. 9: now focuses on the generalized absence of northerly and westerly flow rather than the two cases presented previously.

Fig. S10: now shows corrected wind directions, with Section 5 altered to account for these changes (wind directions are now in better agreement)

Table 2: Previously, wind directions were divided into N/S/E/W categories. This now examines flow divided into mutually-exclusive categories NE/SE/SW/NW with updated values Section 4: Explicit discussion of the bias in MAR for southerly and easterly winds and an extended discussion of northwesterly flow

**7) Explicit comparison of satellite-based, model-based and AWS temperature-based melt occurrence**

We add a comparison of melt occurrence from all satellite measures with AWS temperature based criteria (and associated temperature biases) in order to assess the sensitivity of melt occurrence criteria independently, before addressing the additional impact of wind direction.

Fig. 4: This is a new figure. Fig. 4a uses a meltwater production threshold of 0.4 mm w.e. to detect melt in MAR. A similar figure in Supplemental Fig. S2 (described in #3) Section 3.2 (P10, L21-33) discussed Fig. 4

**8) Justification for the use of the MAR meltwater production threshold of 0.4 mm w.e. for melt occurrence**

The decision to use a threshold of meltwater production exceeding 0.4 mm w.e as a criteria for meltwater occurrence is more thoroughly justified in the context of previous literature as well as via a comparison of melt occurrence estimates both domain-wide and at one AWS station (using satellite-based, model-based and AWS temperature-based estimates for melt occurrence) where multiple MAR thresholds are employed (0.1 mm w.e. -4 mm w.e.).

Data and Methods: P6, L 8-18 Supplemental Fig. S2 Fig. 4

---

## Author Comment (AC4) · 14 Jun 2018

Dear Reviewer,

Please find attached the revised supplementary material with revisions detailed in the first author comment.

Thank you.

Sincerely,

R. Tri Datta

[Figure]

Please also note the supplement to this comment:
https://www.the-cryosphere-discuss.net/tc-2017-253/tc-2017-253-AC4-supplement.pdf

---

## Author Comment (AC5) · 14 Jun 2018

Dear Reviewer,

The major revisions requested are categorized here, whereby each major Revision# is described in the file "MajorRevisions_TC_AP_RDatta", also attached. Our response to minor comments are listed after.

Many thanks for your consideration.

Sincerely,

R. Tri Datta

**Major Revisions:**

**1) Observational uncertainty**
Original comment:
What about observational uncertainty of satellite data or uncertainties introduced by
the postprocessing of satellite data? Would it be possible to include a specific errorestimate
to better evaluate the model results and to take into account the observational
uncertainty?

Response
In addition to more focus in the introduction, greater attention is given to the discussion of the spatial resolution of satellite sources, although the errors associated with the postprocessing of satellite data were difficult to quantify here. This is partially addressed in Revision #7, #8,

**2) Driving reanalysis**
Original Comment:
What about the impact of ERA-Interim as driving reanalysis data? Would it be possible
to add it in the evaluation? Could the mentioned aspects of wind biases and thus
resulting biases of melt occurrence have also their origin in the obtained large-scale
atmospheric information given by the boundary condition?

Response:
This has been addressed in Revision #4

**3) Additional model deficiencies**
Original comment:
Could the mentioned cold bias in MAR (when maximum temperature and average daily
temperature exceed 0 degree Celsius) origin from other model deficiencies as well? So
far only wind is considered.

Response
Other potential causes for melt are highlighted in greater detail in the results        P15 L9-17

The potential effects of horizontal and vertical resolution are also expanded upon (although these do relate to wind flow). This is discussed in Revision #1.

**4) Excessive abbreviations**
Original comment:
In Section 4.2 and 4.2.1 there are many abbreviations introduced which makes it a bit difficult to read. Would it be possible to already introduce those in the methods part and provide a table as overview? Or maybe it is possible to reduce the amount of abbreviations used in the text.
Response:
This now includes an overview table as Table 1

**5) Comparisons with other studies**
Original comment:
It would be interesting to discuss presented results (e.g. the underestimation in melting in the center and east of the Larsen C ice shelf) in greater detail to other studies e.g. to other regional climate model studies over the Antarctic region or in general in terms of e.g. issues in snow melting (e.g. onset and ending) in other regions. Also GCMs might have similar issues that would be of interest to consider.
Response:
We have altered the text to emphasize the relative advantages/disadvantages of hydrostatic vs. non-hydrostatic models and an intermodal comparison with the hydrostatic model RACMO2.3p2 (Revision #2, #5)

**Response to Minor Comments:**

Page 3 l. 29 + l. 33: use same space before unit                   – corrected, P3, L23
Page 4 l. 20: change to föhn                                       Paragraph reordered
Page 5 in section 2.1: Please mention the size of the model domain
                                                        Lat/lon boundaries added in P 5 L25
Page 5 l. 2: explain abbreviation RCM
                        Term now fully introduced in Introduction P3, L14,15
Page 7 l. 6: add space after where                 Paragraph reorganized for clarity
Page 7 l.35: remove space before Wilks                       - corrected, P8, L33
Page 12 l. 34: citation with 2 brackets               - corrected, P15, L12
Page 34 l. 34: remove second brackets (assuming related to the previous comment)
Page 18 l. 8: remove slash in Royal                        - corrected, P21, L19
Page 24 l. 5: add space before Greenland                          - corrected
Figures:
Fig. 1: Please add coordinates to the axes                               – added
Fig. 2: Please add coordinates to the axes                   – added, Now Fig. 5
Fig: 3: Please have a consistent labeling of axes throughout all the figures 1-8; variable [unit]                                                 - corrected

Fig: 4: Same as Fig. 4                                              - corrected

Fig. 7: ended with a comma ] – corrected, Figure now combined with previous figure.

R

---

## Author Comment (AC8) · 15 Jun 2018

[Figure]

**(a)**

| Step | Effect on Variables |
|---|---|
| **1) Energy from Rain**
Energy <- RI * Cw * $T_{exc}$ * t | Energy    + or - |
| **2) Surficial Water Exists?** (a function of $T_{exc}$) | |
| **3) Energy from Snow**
Energy <- ρ * Cs * $T_{exc}$ * ds | Energy    + or - |
| **4) Water from snowpack**
RI <- ρ*ds * (soil humidity)   ρ reduced by soil humidity | Rain Intensity  +
density - |
| **5) MELT (when Energy is positive)**
melted snow <- Energy / (Lf * ρ)
ds -> melted snow -> RI | Energy -
snow depth -
Rain Intensity + |
| **6) Alter the snow history based on whether melting is occurring**
faceted crystals? Liquid water with no faceted crystals?
Liquid water with faceted crystals before? | |
| **7) FREEZE (when Energy is negative)**
Energy / (Lf * ρ) <- frozen water <-  RI
Tsnow -> Energy / (ρ * ds * Cs )
ρ increased by the addition of frozen water | Temperature -
Rain Intensity -
Energy +
Density + |
| **8) Water saturation in snow**
an irreducible portion of the snowpack must contain water.
RI -> irreducible  water in snowpack (constant * pore volume * ds * density(water)
irreducible water in snowpack -> ρ
**\*\*\*** | Rain Intensity -
Density + |

| Step | Effect on Variables |
|---|---|
| **9) Pore hole close off  / superimposed ice**
Whether a pore hole closes off is determined as a function of density, density of
ice and a constant value for pore hole close off density.
Pore close off -> RI converted to surficial water
No pore close off -> RI remains in RI
**\*\*\*** | Surficial Water, Rain Intensity + / - |
| **10) Surficial water runoff**
Final Energy reduced by Energy from rain (Step 1)
A decay function determines the portion of surficial water converted to runoff
Reference: Zuo and Oerlemans, 1996
**\*\*\*** | Surficial water -
Runoff + |
| **11) Conversion back to rain**
Where no superimposed ice occurs (Step 9 above), surficial water is
added back into RI (rain intensity) | Rain Intensity +
Surficial water - |
| **12) Slush**
Where surficial water exists (step 2),  the highest snow/ice layer will
fill the pore volume with water from surficial water, adding to density
slush + <-  surficial water          slush -> ρ +
**\*\*\*** | Surficial water -
density + |
| **13) Add/Subtract   Deposition/Sublimation**
Snowpack either +/ -
 DepOrSubl= t * LHF /  (Lx * ρ) <- -> ds       added to the snowpack
Energy of vapor calculated
EnVp = (Cs * $T_{exc}$-Lf * (1-soil humidity) )   /  (1 + (DepOrSubl / ds))

          and used to alter humidity of soil/snow
Hum = 1 + (EnVp – Texc * Cs) / Lf
          as well as the temperature
$T_{exc}$ = (EnVp + Lf*(1-Hum) )/ Cs | Soil/Ice Humidity +/-
Temp +/-
snowpack    +/- |

**\*\*\*** denotes steps where tuning is possible or separate physics are calculated depending on the region

| Variables | | | Constants: | | |
|---|---|---|---|---|---|
| RI | Rain Intensity | kg m$^{-2}$ s$^{-1}$ | Cw | Heat Capacity of Water | Jkg$^{-1}$K$^{-1}$ |
| $T_{exc}$ | Temp above/below 0 | C | Cs | Heat Capacity of snow | Jkg$^{-1}$K$^{-1}$ |
| t | Time elapsed | s | Lf | Latent heat of fusion | Jkg$^{-1}$ |
| ρ | density snow | kg m$^{-3}$ | Lx | Latent Heat of Vap/Subl | Jkg$^{-1}$ |
| ds | snow depth | m | | | |
| LHF | latent heat flux  W m$^{-2}$ | | | | |

**(b)**

**Figure S1: Diagram (a)  and description (b) of the physical  processes within MAR's SISVAT (Soil Ice Vegetation Atmosphere Transfer Scheme) calculating meltwater production and meltwater percolation into the snowpack from the energy balance and the presence of water, using the density of the snowpack (ρ), temperature of the surface boundary layer (TSBL) and temperature of the snow (TSNOW).**

[Figure]

Figure S2: Average MAR melt duration from 2000-2009 using different thresholds for total daily meltwater production to determine melt occurrence (1) 0.1 mm w.e. (b) 0.4 mm w.e. (c) 1 mm w.e. (d) 4 mm w.e. Major differences in average yearly melt duration between melt thresholds (e) 0.1 mm w.e. – 0.4 mm w.e. (f) 1 mm w.e. – 0.4 mm w.e. (g) 4 mm w.e. – 1 mm w.e.

[Figure]

5      (a)                  (b)                 (c)                 (d)                 (e)                 (f)

**Figure S3: Average Melt Onset date from multiple sources (a) MAR, Liquid Water Content > 0.4% for three consecutive days. (b) MAR Total Melt Flux > 0.4 mmwe for 1 day or more. Satellite-based: (c) PMW 240 algorithm (d) PMW ALA (e) PMW Zwa (f) QuikSCAT. Day shown is the first day of a sustained three-day melt period for satellite estimates as well as LWC$_{1m}$, Date number is defined beginning in Jan 1$^{st}$. of year1, such that 365 represents Dec 31$^{st}$ of year1. All averages are taken from the 2000-2009 period to retain consistency with the availability of QuikSCAT data.**

[Figure]

14              (a)                              (b)                              (c)

**Figure S4: Number of 10km MAR grid cells from the NE basin (y axis) showing the avg number of total melt days (2001-2014) from three passive microwave algorithms: (a) PMW 240  (b) PMW ALA (c) PMW zwa**

| Region | Avg. Annual Melt Days (2001-2014) [Days] | Elevation [m] | Avg coincident MAR Meltwater Production NDJF (2001 to 2014) [mmWE/100km$^2$] |
|---|---|---|---|
| **240 L** | $1 \leq D < 10$ | $833.70 \pm 539.62$ | 7.81 |
| **240 M** | $10 \leq D < 30$ | $72.37 \pm 90.98$ | 55.32 |
| **240 H** | $30 \leq D$ | $42.94 \pm 17.78$ | 95.09 |
| **ALA L** | $1 \leq D < 15$ | $1016.13 \pm 525.80$ | 7.28 |
| **ALA M** | $15 \leq D < 40$ | $125.97 \pm 200.67$ | 62.94 |
| **ALA H** | $40 \leq D$ | $56.92 \pm 56.69$ | 128.72 |
| **zwa L** | $1 \leq D < 20$ | $1165.99 \pm 513.24$ | 7.82 |
| **zwa M** | $20 \leq D < 45$ | $374.80 \pm 471.47$ | 47.55 |
| **zwa H** | $45 \leq D$ | $101.73 \pm 173.27$ | 126.19 |
| **CL Region** | | $42.67 \pm 17.68$ | |
| PMW All | $36.63 \pm 4.01$ | $39.15 \pm 17.87$ | 96.15 |
| MF$_{0.4}$ | $21.29 \pm 9.10$ | $42.15 \pm 16.05$ | 143.08 |
| **NL Region** | | $594.12 \pm 601.20$ | |
| PMW All | $7.74 \pm 8.90$ | $86.72 \pm 137.87$ | 41.24 |
| MF$_{0.4}$ | $26.68 \pm 24.94$ | $126.88 \pm 159.87$ | 231.97 |

**Table S1: Average statistics for regions of melt occurrence, restricted to the NE basin. The first 9 rows indicate regions where melt occurrence is determined by a PMW algorithm (i.e. 240) restricted by the number of days where melt occurrence (i.e. 240 L, where the number of avg annual melt days is between 1 and 10). CL and NL regions are described in text. Row indicating "PMW All"or "MF$_{0.4}$" in left column  implies that corresponding statistics in columns 2-4 are calculated for where melt occurrence meets these conditions**

[Figure]

4    **Figure S5: Avg Melt Days (2001-2014) from three passive microwave algorithms (described in text). Green shows PMW**
5    **240. Blue shows PMW ALA. Red shows PMW zwa.**

[Figure]

Figure S6: Maximum depth of MAR-modeled meltwater percolation (MAR) into the snowpack over the melt season. Colors indicate the percentage of grid cells where meltwater reaches the corresponding maximum depth (y axis) for the month (x axis), such that each column per month totals to 100%. Maximum percolation depth is determined by the maximum depth over the month where liquid water content in MAR is greater than 0.02 kg/kg. Grid cells for each column are restricted to the corresponding month during the 2001-2014 period which fulfill the conditions (a) 240-L (b) ALA-L (c) zwa-L (d)240-M (e) ALA-M (f) zwa-M (g) 240-H (h) ALA-H (i) zwa-H, as defined in table 1.

[Figure]

3 AWS14/Larsen

4           (a)              (b)           (c)

5 AWS15

6           (d)              (e)           (f)

8 **Figure S7: Left column is for surface pressure. Middle column for daily-averaged 2m air temperature, right column for**
9 **2m wind speed. Stations are as follows: (a)(b)(c) AWS 14/Larsen Ice Shelf, which are co-located in MAR (d)(e)(f) AWS 15**

[Figure]

2
(a)                                                            (b)

4
(c)                                                            (d)

6
7
8
9   **Figure S8: Seasonal Avg Ts climatology for spring (SON) and summer(DJF) with envelope indicating one standard**
10  **deviation, Red: computed for available data from AWS station, with quality control as described in section 2. Green:**
11  **MAR daily-averaged  T2m data restricted to AWS-data availability. Blue: MAR daily-averaged T2m data for the full**
12  **period (1999-2014). Data is shown for (a)(b) AWS 14 (c)(d) Larsen Ice Shelf  (e)(f) AWS 15**

|  | 01-02 | 02-03 | 03-04 | 04-05 | 05-06 | 06-07 | 07-08 | 08-09 | 09-10 | 10-11 | 11-12 | 12-13 | 13-14 |
|---|---|---|---|---|---|---|---|---|---|---|---|---|---|
| AWS14 |  |  |  |  |  |  |  |  | 0.98 | 0.98 | 0.98 | 0.99 | 0.97 |
| AWS15 |  |  |  |  |  |  |  |  | 0.98 | 0.98 |  | 0.60 | 0.97 |
| Larsen IS | 0.99 | 0.98 | 0.98 |  |  |  | 0.98 | 0.98 | 0.98 | 0.98 | 0.98 |  |  |

**Table S2: R² values between MAR and AWS data for Surface Pressure in Summer (DJF) for years shown**

|  | 01-02 | 02-03 | 03-04 | 04-05 | 05-06 | 06-07 | 07-08 | 08-09 | 09-10 | 10-11 | 11-12 | 12-13 | 13-14 |
|---|---|---|---|---|---|---|---|---|---|---|---|---|---|
| AWS14 |  |  |  |  |  |  |  |  | 2.36 | 1.71 | 2.36 | 2.4 | 2.48 |
| AWS15 |  |  |  |  |  |  |  |  | 2.46 | 2.05 |  | 6.04 | 1.97 |
| Larsen IS | 1.14 | 1.71 | 2.01 |  |  |  | 2.07 | 2.25 | 2.25 | 1.77 | 2.19 |  |  |

**Table S3: Root Mean Squared Error between MAR and AWS data for Pressure [hPa] in Summer (DJF) for years shown**

|  | 01-02 | 02-03 | 03-04 | 04-05 | 05-06 | 06-07 | 07-08 | 08-09 | 09-10 | 10-11 | 11-12 | 12-13 | 13-14 |
|---|---|---|---|---|---|---|---|---|---|---|---|---|---|
| AWS14 |  |  |  |  |  |  |  |  | -2.10 | -1.32 | -2.04 | -2.26 | -2.17 |
| AWS15 |  |  |  |  |  |  |  |  | -2.21 | -1.72 |  | -0.84 | -1.59 |
| Larsen IS | -0.65 | -1.21 | -1.66 |  |  |  | -1.59 | -1.82 | -1.95 | -1.36 | -1.83 |  |  |

**Table S4: Mean Error (MAR-AWS) for Pressure [hPa] in Summer (DJF) for years shown**

|  | 01-02 | 02-03 | 03-04 | 04-05 | 05-06 | 06-07 | 07-08 | 08-09 | 09-10 | 10-11 | 11-12 | 12-13 | 13-14 |
|---|---|---|---|---|---|---|---|---|---|---|---|---|---|
| AWS14 |  |  |  |  |  |  |  |  | 0.68 | 0.41 | 0.71 | 0.39 | 0.59 |
| AWS15 |  |  |  |  |  |  |  |  | 0.66 | 0.50 | 0.63 | 0.35 | 0.58 |
| Larsen IS | 0.36 | 0.27 | 0.56 |  |  |  | 0.57 | 0.67 | 0.70 | 0.44 | 0.71 |  |  |

**Table S5: R² values between MAR and AWS data for daily-averaged 2m air temperature in Summer (DJF) for years shown**

|  | 01-02 | 02-03 | 03-04 | 04-05 | 05-06 | 06-07 | 07-08 | 08-09 | 09-10 | 10-11 | 11-12 | 12-13 | 13-14 |
|---|---|---|---|---|---|---|---|---|---|---|---|---|---|
| AWS14 |  |  |  |  |  |  |  |  | 1.91 | 2.19 | 2.56 | 2.60 | 2.47 |
| AWS15 |  |  |  |  |  |  |  |  | 2.10 | 1.99 | 2.91 | 2.72 | 2.45 |
| Larsen IS | 1.39 | 2.20 | 3.37 |  |  |  | 1.82 | 2.36 | 1.62 | 1.81 | 2.34 |  |  |

**Table S6: Root Mean Squared Error between MAR and AWS data for daily-averaged 2m air temperature [°C] in Summer (DJF) for years shown**

|  | 01-02 | 02-03 | 03-04 | 04-05 | 05-06 | 06-07 | 07-08 | 08-09 | 09-10 | 10-11 | 11-12 | 12-13 | 13-14 |
|---|---|---|---|---|---|---|---|---|---|---|---|---|---|
| AWS14 |  |  |  |  |  |  |  |  | 0.98 | 1.00 | 1.61 | 0.32 | 1.30 |
| AWS15 |  |  |  |  |  |  |  |  | 1.06 | 0.81 | 1.60 | 0.32 | 1.11 |
| Larsen IS | -0.52 | 0.15 | 2.22 |  |  |  | 0.94 | 1.23 | 0.36 | 0.26 | 1.01 |  |  |

**Table S7: Mean Error (MAR-AWS) for daily-averaged 2m air temperature [°C] in Summer (DJF) for years shown**

[Figure]

[Figure]
 270°-90° recorded by AWS   90-270° recorded by MAR

| | Pr. [%] | Mean Avg T2m [°C] | | Mean MaxT2m [°C] | | Temp Bias [°C] | | | |
|---|---|---|---|---|---|---|---|---|---|
| | | MAR | AWS | MAR | AWS | AvgT2m | AvgT2m >0 | MaxT2m | MaxT2m >0 |
| ALL | 8 | -2.31 (±2.27) | -3.18 (±3.61) | -1.50 (±2.19) | 0.54 (±3.25) | 0.88 (±2.87) | -1.45 (±1.56) | -2.07 (±2.88) | -3.36 (±2.23) |
| MAR | 8 | -0.70 (±0.80) | -1.89 (±2.69) | -0.27 (±1.32) | 1.73 (±2.74) | 1.19 (±2.53) | -0.89 (±1.15) | -2.00 2.85(±) | -3.08 (±2.28) |
| PMWEx | 6 | -1.76 (±1.16) | -1.73 (±1.63) | -0.90 (±1.17) | 1.45 (±2.19) | -0.04 (±1.46) | -2.06 (±0.71) | -2.35 (±2.12) | -2.99 (±1.99) |
| QSEx | 6 | -1.91 (±1.18) | -2.04 (±2.21) | -0.97 (±1.29) | 1.25 (±2.38) | 0.12 (±2.03) | -2.81 (±1.60) | -2.28 (±2.39) | -3.15 (±2.15) |

**Table S8: Temp averages and biases, proportions where AWS-observed northerly winds are reported as southerly in MAR, as a percentage of all wind direction values for the condition. Conditions for ALL , PMWEx and QSEx (in text)**

[Figure]

[Figure]
 180-360° recorded by AWS   0-180° recorded by MAR

| | Pr. [%] | Mean Avg T2m [°C] | | Mean MaxT2m [°C] | | Temp Bias [°C] | | | |
|---|---|---|---|---|---|---|---|---|---|
| | | MAR | AWS | MAR | AWS | AvgT2m | AvgT2m >0 | MaxT2m | MaxT2m >0 |
| ALL | 10 | -2.25 (±2.24) | -3.35 (±3.44) | -1.50 (±2.16) | 0.35 (±3.96) | 1.09 (±2.32) | -1.52 (±1.41) | -1.88 (±3.24) | -3.76 (±2.68) |
| MAR | 10 | -0.64 (±0.74) | -1.40 (±1.84) | -0.16 (±1.33) | 2.12 (±2.86) | 0.76 (±1.75) | -1.07 (±1.05) | -2.24 (±2.63) | -3.19 (±2.19) |
| PMWEx | 8 | -1.92 (±0.99) | -1.88 (±1.95) | -1.23 (±1.24) | 0.86 (±3.82) | -0.04 (±1.60) | -2.60 (±1.10) | -2.09 (±3.29) | -3.86 (±2.95) |
| QSEx | 9 | -2.12 (±1.26) | -2.83 (±2.63) | -1.23 (±1.24) | 0.86 (±3.82) | 0.71 (±2.18) | -2.85 (±1.52) | -2.09 (±3.29) | -3.83 (±2.59) |

**Table S9: Temp averages and biases, proportions where AWS-observed westerly winds are reported as easterly in MAR, as a percentage of all wind direction values for the condition. Conditions for ALL , PMWEx and QSEx (in text)**

2
3

 180-270° recorded by AWS  180-270° recorded by MAR
[Figure]

|  | Pr. [%] | Mean Avg T2m [°C] | | Mean MaxT2m [°C] | | Temp Bias [°C] | | | |
|---|---|---|---|---|---|---|---|---|---|
|  |  | MAR | AWS | MAR | AWS | AvgT2m | AvgT2m >0 | MaxT2m | MaxT2m >0 |
| ALL | 7 | -2.44 (±2.41) | -3.41 (±3.56) | -1.69 (±2.55) | -0.95 (±3.56) | 0.97 ( ±1.88) | -1.14 (±1.14) | -0.76 (±2.04) | -1.83 ( ±1.57) |
| MAR | 6 | -0.33 ( ±0.60) | -1.06 (±1.35) | 0.25 (±1.37) | 0.99 (±1.74) | 0.73 (±1.11) | -0.54 ( ±0.82) | -0.71 (±1.48) | -1.17 ( ±1.31) |
| PMWEx | 6 | -1.55 (±1.07) | -1.97 (±1.95) | -0.63 (±1.30) | 0.97 (±2.20) | 0.41 (±1.60) | -1.57 (±0.15) | -1.60 (±1.91) | -2.10 (±1.52) |
| QSEx | 7 | -2.01 (±1.30) | -2.50 (±2.18) | -1.10 (±1.48) | 0.22 (±2.50) | 0.49 (±1.58) | -2.23 (±0.76) | -1.33 (±2.07) | -2.33 (±1.59) |

**Table S10: Temp averages and biases, proportions where AWS-observed southesterly winds are preserved in MAR, as a percentage of all wind direction values for the condition. Conditions for ALL , PMWEx and QSEx (in text)**

8

 180-270° recorded by AWS  90-180° recorded by MAR

|  | Pr. [%] | Mean Avg T2m [°C] | | Mean MaxT2m [°C] | | Temp Bias [°C] | | | |
|---|---|---|---|---|---|---|---|---|---|
|  |  | MAR | AWS | MAR | AWS | AvgT2m | AvgT2m >0 | MaxT2m | MaxT2m >0 |
| ALL | 4 | -2.13 (±1.77) | -3.12 (±3.14) | -1.71 (±1.69) | -1.01 (±3.39) | 0.99 ( ±2.01) | -1.27 (±0.59) | -0.74 (±2.51) | -2.81 (±1.42) |
| MAR | 4 | -0.77 ( ±0.40) | -1.27 (±1.20) | -0.54 (±0.56) | 1.35 (±1.98) | 0.49 (±1.30) | -1.21 (±0.57) | -1.84 (±2.22) | -2.92 (±1.59) |
| PMWEx | 3 | -1.69 (±0.62) | -1.87 (±1.95) | -1.27 (±0.73) | -0.32 (±1.62) | 0.18 (±1.48) | N/A | -0.95 (±1.26) | -1.98 (±0.56) |
| QSEx | 4 | -1.90 (±0.79) | -2.99 (±2.59) | -1.47 (±0.77) | -1.28 (±2.81) | 1.08 (±2.13) | N/A | -0.16 (±2.39) | -2.35 (±0.94) |

**Table S11: Temp averages and biases, proportions where AWS-observed southwesterly winds are reported as southeasterly in MAR, as a percentage of all wind direction values for the condition. Conditions for ALL , PMWEx and QSEx (in text)**

[Figure]
 270-360° recorded by AWS    0-270° recorded by MAR
[Figure]

| | Pr. [%] | Mean Avg T2m [°C] | | Mean MaxT2m [°C] | | Temp Bias [°C] | | | |
|---|---|---|---|---|---|---|---|---|---|
| | | MAR | AWS | MAR | AWS | AvgT2m | AvgT2m >0 | MaxT2m | MaxT2m >0 |
| ALL | 3 | -2.74 (±3.11) | -3.78 (±4.45) | -1.19 (±3.08) | 2.46 (±4.54) | 1.04 ( ±3.35) | -1.71 (±2.08) | -3.72 (±4.15) | -4.98 (±3.61) |
| MAR | 3 | -0.61 (±1.34) | -2.01 (±4.10) | 0.40 (±2.38) | 3.31 (±4.19) | 1.40 (±3.72) | -0.82 (±1.41) | -2.91 (±3.77) | -4.04 (±3.08) |
| PMWEx | 3 | -2.51 (±1.69) | -2.43 (±2.24) | -0.76 (±1.68) | 4.60 (±4.20) | -0.08 (±1.92) | -2.89 (±1.68) | -5.36 (±3.80) | -5.57 (±3.80) |
| QSEx | 4 | -2.71 (±1.82) | -3.09 (±2.92) | -1.00 (±1.92) | 3.23 (±4.01) | 0.38 (±2.65) | -3.42 (±2.12) | -4.34 (±3.69) | -5.11 (±3.46) |

**Table S12: Temp averages and biases, proportions where AWS-observed southwesterly winds are reported as southeasterly in MAR, as a percentage of all wind direction values for the condition. Conditions for ALL, PMWEx and QSEx (in text)**

[Figure]

(a)  CL melt extent            (b)  NL melt extent

(c)                            (d)

(e)                            (f)

GridCells

Figure S9: Melt extent ( from satellite and MAR)  and temperature (from AWS and MAR)  over the 2001-2002 melt season., (a)  CL region melt extent (b) NL region melt extent. Masks described in text and shown in inset of Fig.2. (c)(d) raw QuikSCAT backscatter for the number of QuikSCAT grid cells (~5 km$^2$) where both MAR and QuikSCAT ft3 detect melt (e)(f) raw QuikSCAT backscatter for the number of QuikSCAT grid cells where the QuikSCAT ft3 algorithm detects melt, but MAR does not.

[Figure]

2      (a) AWS Nov      (b) AWS Dec      (c) AWS Jan      (d) AWS Feb

4      (e) MAR Nov      (f) MAR Dec      (g) MAR Jan      (h) MAR Feb

8           (i)

10 **Figure S3 : Wind roses shown for the Larsen IS AWS station in 2001-2002 for (a)Nov (b)Dec (c) Jan (d) Feb. Wind roses**
11 **shown for the MAR grid cell co-located to the Larsen IS AWS station in 2001-2002 for (e) Nov (f)Dec (g) Jan (h) Feb.(i)**
12 **Proportion of wind direction (directions shown in inset) for all grid cells where MAR melt occurs in the NL region over the**
13 **melt season**

[Figure]

**Figure S11: AWS and MAR AvgTs and MaxTs for the 2001-2002 season (Larsen Ice Shelf AWS statio**

[Figure]

Figure S4: Melt occurrence using the $MF_{0.4}$ criteria (a,b,c) and meltwater production (d,e,f) from Nov 1, 2004 – Feb 28, 2005 for MAR v3.9 at (a,d) 10km and 23 sigma layers (b,e) 10km and 32 sigma layers and (c,f) 5km horizontal spatial resolution and 23 sigma layers

[Figure]

3
4
    (a) MAR v3.9 5km      (b) MAR v3.9 10km      (c) MAR v3.9 10km V

6
    (d) MAR v3.5.2 10km      (e) AWS

8  **Figure S5: Wind roses at the Larsen IS AWS location (shown in Fig. 1) for Nov 1, 2004 – March 31, 2005 for MAR 3.9 at a 5km resolution, 23 sigma layers, daily values**
9  **(a) 10 km and 23 sigma layers, daily values (b) 10km resolution and 32 sigma layers, daily values (c), MAR 3.5.2 at a 10km resolution and 23 sigma layers, daily values (d)**
10  **and AWS at 3-hourly values (e). MAR values are calculated only when AWS data is available and AWS data reports no values between Dec 20, 2004 and February 12,**
11  **2005**

---

## Author Response (AR1)

In addition to a change in the title, there are 9 major changes in response to author comments. The 10[th] change is in response to a bug in the analysis which we discovered in the process of revisions. Additional changes are made to language for the purpose of clarity throughout (especially the abstract) and several additional references have been included in light of developments in the subject area in the last few months.

**MAJOR REVISIONS**

**1) Model Choice.**

Comments from Referee #1
My main concern is that the focus of the paper is to evaluate the MAR model in terms of melting, however there appears to be relatively little discussion on the impact of the chosen model options, such as the horizontal and vertical resolution, which limits the readers understanding of how efficient MAR is at reproducing melt on the ice shelf, as seen in satellite observations. The potential impact of model choice, and model physics is most clear in the results and discussion of the wind direction, where there is a large discrepancy between the model and observations.
.
.
The above comment is also linked to the relatively coarse (for this region and this topic) horizontal resolution used here. Previous föhn studies use much finer resolution (5km, 1.5km) and suggest that this resolution is required for adequate föhn representation. Authors discuss the Van Wessem et al (2015) study which suggests higher resolution than 5.5km. However the authors appear to use the following statement: "where hydro- static assumption is preserved (such as this model run), higher resolutions may inhibit flow in the model. . ." (pg 3, line 35/36) to justify using a lower horizontal resolution. To address this issue, I think a sensitivity study using higher resolution is required. It doesn't have to be for the full-time period, but should capture at least a season of melt to assess whether the spatial resolution could improve the results, and whether the breakdown of the hydrostatic equation does limit air flow. You should also add more to the discussion about this. The spatial resolution is not mentioned at all in the discussion, and as found in other papers (Van Wessem et al 2015, Turton et al 2017, Elvidge et al 2015), higher resolution runs do capture föhn winds.

The abstract states that increased spatial resolution and topographic resolution could improve the output from MAR, but there is no mention of this in the discussion or results. You should not include statements in the abstract which do not reflect the results of the study. Either address whether changing the spatial or topographical resolution does impact the modelled melt or near-surface conditions, or remove this from the abstract.

The vertical resolution of MAR's atmosphere is very coarse, especially to have the lowest model level at 2m above the surface (pg 5, lines 10 and pg7 line 29/30). What is the vertical discretisation of your levels? The WRF model for instance has difficulties if there is over 1km between model levels, or if the stretching factor is greater than 20%. Similar to the first major comment, a sensitivity study is required to assess the impact of this vertical resolution on the representation of the near-surface conditions and the wind. This is much coarser than many

studies of this kind and studies using MAR (see for example, Gallee et al 2015 or Wyard et al 2016 who both use 60 vertical levels). Again, this doesn't need to be the full period (and shouldn't be as this would be a huge/long undertaking) but a full season should be tested using a number of higher vertical levels.

Author Response:
We now include a comparison between higher-resolution runs of a newer version MAR over the 2004-2005 melt season. The comparison includes 3 versions of MAR: (a) at 10km (b) where the horizontal resolution is increased from 10km to 5km and (c) where the vertical discretization is increased from 23 to 32 sigma layers. The variables examined are meltwater production, melt occurrence (over the domain) and wind speed/direction at the Larsen Ice Shelf AWS location. We find that an increase in resolution limits melt over the Larsen C ice shelf and increases southeasterly flow, suggesting that while the hydrostatic assumption is kept, the effect of increased resolution will lead to reduced melt overall, but potentially enhance the accuracy of melt just east of the AP due to better-resolved topography. This is specifically presented in the Abstract, discussed in detail in the Introduction and presented in the Discussion and Conclusions.

Author's Changes in the Manuscript
Abstract: P1, L29-31
Introduction: P3, L 4-12, 26-36
Data and Methods: P5, L 30-37
Discussion and Conclusion: P17, L30 – P18, L9
Supplement Fig. S12, S13

**2) The hydrostatic assumption**

Comments from Referee #1

The hydrostatic assumption and horizontal resolution of MAR. In the abstract, authors state that "melting in the East AP can be initiated by both sporadic westerly föhn flow over the AP and by northerly winds advecting warm air from lower latitudes. To assess MAR's ability to simulate these physical processes, this study. . ." (line Pg 1, 24- 27). Then later in the discussion you state that MAR can't accurately represent the wind direction and föhn processes due to the model's hydrostatic assumption, and state that a non-hydrostatic model would do better (pg 15 line 23-27). Models which have previously, successfully captured the föhn characteristics (WRF and UM) are nonhydrostatic, and this appears to be known to the authors prior to the study as in the introduction (Pg3, line 33), they discuss the (non-hydrostatic) RACMO study, and justify their use of a coarser resolution due to the hydrostatic assumption. If a large part of the study is to assess the impact of wind on the melting, why chose a model which can't represent the dominant westerly flow (and subsequent downward föhn flow) over the AP? If an objective of this study was to attempt to model this type of flow using a model with hydrostatic assumption, then this should be made clearer, and authors should note any previous studies of this kind.

Comments from Referee #2:
It would be interesting to discuss presented results (e.g. the underestimation in melting in the center and east of the Larsen C ice shelf) in greater detail to other studies e.g. to

other regional climate model studies over the Antarctic region or in general in terms of e.g. issues in snow melting (e.g. onset and ending) in other regions. Also GCMs might have similar issues that would be of interest to consider

Author Response:
We have altered the text to emphasize the relative advantages/disadvantages of hydrostatic vs. non-hydrostatic versions of the model, i.e. the accuracy of winds (in non-hydrostatic models such as WRF) vs. the long run periods and sophistication of the snowpack (in hydrostatic models such as MAR or RACMO2.3p2).

We now include a more thorough review of recent non-hydrostatic modeling studies (King et al., 2017; Turton et al., 2018; Bozkurt et al., 2018) noting that factors other than föhn melt are important over the Larsen C ice shelf as well as recent work showing that even a high-resolution non-hydrostatic model was not fully able to resolve föhn characteristics.

We have corrected a typo mis-stating that RACMO2.3p2 is a non-hydrostatic model, added greater detail about recent publications with RACMO2.3p2 over the AP, and included a direct comparison of melt occurrence/meltwater production between RACMO and MAR. References to recent work on RACMO3.2p2 over the AP are included (Van Wessem et al., 2018; Weisenekker et al., 2018)

Author's Changes in the Manuscript
Re: non-hydrostatic models:
Introduction: P3, L 2-12
Discussion and Conclusion: P18, L10-29

Re: hydrostatic models (RACMO2.3p2):
Introduction: P3, 26-36, P4, 7-9
Data and Methods: P 6, 19-23
Results: P10, L 3-19; P11, L18-21
Fig. 3, Fig. 5c

**3) Overemphasis föhn winds**

Comments from Referee #1:
In the abstract and introduction, a fair amount of emphasis is put on the role of föhn winds and northwesterly winds e.g (Pg 1, line 25, 32, 33, 35, Pg 3, line 5-21, 34-37, Pg 4, line 1, 20). However, in the results and discussion, this is not discussed thoroughly, either in the context of other studies, or how well MAR can model these features. More discussion of the föhn characteristics and melt related effects needs to be included in your discussion to have such a prevalence in the earlier sections.

Author Response:
The original discussion about föhn winds has been substantially limited to where the main emphasis is placed on previous studies with a non-hydrostatic model. More emphasis is placed

here on the distinction between the initial intrusion of föhn flow (which high-resolution hydrostatic models may capture) vs. the eastward propagation towards the edge of the Larsen C ice shelf (where the comparison with AWS stations are conducted).
However, a section on northwesterly flow biases is specifically included to address the effects of probably föhn flow

Author's Changes in the Manuscript
Introduction: P3, L 1-12
Results: P16, L7-20
Discussion and Conclusions: P18, L 18-20

**4) Length of the model run**

Comments from Referee #1:
An additional novel aspect which this paper does not mention is the length of this modelling study. Output for the model are for 15 years, which is a long study period over this region. Previous melt-föhn studies have largely focused on case studies, or shorter time periods (e.g Elvidge et al 2015, Grosvenor et al (2014), King et al 2017). More emphasis could be put on the length of study, as this is of importance.

Author Response:
The length of the study has been emphasized in several places.

Author's Changes in the Manuscript
Abstract: P1, L18-20
Introduction: P4, 1-2
Discussion and Conclusions: P19,L1-5

**5) Driving Reanalysis**

Comments from Referee #2:

What about the impact of ERA-Interim as driving reanalysis data? Would it be possible to add it in the evaluation? Could the mentioned aspects of wind biases and thus resulting biases of melt occurrence have also their origin in the obtained large-scale atmospheric information given by the boundary condition?

Author Response:
To understand the impact of forcing on the representation of wind dynamics in MAR, we have included a comparison of ERA-Interim wind fields and discussed possible reasons for the differences.

Author's Changes in the Manuscript
Fig. 8 d,k
Data and Methods: P5, L 26-28
Results: P14, L35 – P15 7

**6) Explicit comparison of satellite-based, model-based and AWS temperature-based melt occurrence**

Comments from Referee #2:
What about observational uncertainty of satellite data or uncertainties introduced by the postprocessing of satellite data? Would it be possible to include a specific errorestimate to better evaluate the model results and to take into account the observational uncertainty?

Author Response
In addition to more focus in the introduction, greater attention is given to the discussion of the spatial resolution of satellite sources, although the errors associated with the postprocessing of satellite data were difficult to quantify here. Additionally, we add a comparison of melt occurrence from all satellite measures with AWS temperature based criteria (and associated temperature biases) in order to assess the sensitivity of melt occurrence criteria independently, before addressing the additional impact of wind direction.

Author's Changes in the Manuscript
Fig. 4: This is a new figure. Fig. 4a uses a meltwater production threshold of 0.4 mm w.e. to detect melt in MAR. A similar figure in Supplemental Fig. S2 (described in #3)
Section 3.2 (P10, L21-33) discussed Fig. 4

**7) Justification for the use of the MAR meltwater production threshold of 0.4 mm w.e. for melt occurrence**

Comments from Referee #2:
What about observational uncertainty of satellite data or uncertainties introduced by the postprocessing of satellite data? Would it be possible to include a specific errorestimate to better evaluate the model results and to take into account the observational uncertainty?

Author Response
The decision to use a threshold of meltwater production exceeding 0.4 mm w.e as a criteria for meltwater occurrence is more thoroughly justified in the context of previous literature as well as via a comparison of melt occurrence estimates both domain-wide and at one AWS station (using satellite-based, model-based and AWS temperature-based estimates for melt occurrence) where multiple MAR thresholds are employed (0.1 mm w.e. – 4 mm w.e.).

Author's Changes in the Manuscript
Data and Methods: P6, L 8-18
Supplemental Fig. S2
Fig. 4

**8) Additional model deficiences**

Could the mentioned cold bias in MAR (when maximum temperature and average daily temperature exceed 0 degree Celsius) origin from other model deficiencies as well? So far only wind is considered.

Author Response
Other potential causes for melt are highlighted in greater detail in the results
The potential effects of horizontal and vertical resolution are also expanded upon (although these do relate to wind flow). This is discussed in Revision #1.

Author's Changes in the Manuscript:
P15 L9-17

**9) Excessive Abbreviations**

Comments from Reviewer #2:
In Section 4.2 and 4.2.1 there are many abbreviations introduced which makes it a bit difficult to read. Would it be possible to already introduce those in the methods part and provide a table as overview? Or maybe it is possible to reduce the amount of abbreviations used in the text.

Author's Changes in the Manuscript
This now includes an overview table as Table 1

**10)        Computation of Wind Direction**

In the process of addressing the major revisions, we discovered a bug with the computation of wind direction in MAR. The now-corrected computation of wind direction substantially reduced the wind direction biases, although we address the biases that are present in light of how the absence of westerly flow affects melt on the eastern Larsen C ice shelf (where the AWSs are located).

Author's Changes in the Manuscript
Fig. 9: now focuses on the generalized absence of northerly and westerly flow rather than the two cases presented previously.
Fig. S10: now shows corrected wind directions, with Section 5 altered to account for these changes (wind directions are now in better agreement)
Table 2: Previously, wind directions were divided into N/S/E/W categories. This now examines flow divided into mutually-exclusive categories NE/SE/SW/NW with updated values
Section 4: Explicit discussion of the bias in MAR for southerly and easterly winds and an extended discussion of northwesterly flow

**Response to Minor Comments (Referee #1):**

**Abstract:**
Line 34: Authors state that reducing the underestimation of flow may be obtained by increasing the spatial resolution, but this is not given much discussion later in the paper. Either remove it and focus on hydrostatic assumption, or include changes to the spatial resolution in the discussion- either results from the suggested sensitivity study, or by discussing other studies.          *See Response to Major Concerns above, #1*

Line 35: You mention reducing the underestimation of flow may be obtained by using higher-resolution topography, but this is not mentioned anywhere else in the paper. Similarly, you do not state what topography is used in the model, or what resolution it is.          *See Response to Major Concerns above, #1*
          *Topography shown in P5, L21-22*

**Introduction:**
Pg 2 Line 7: remove 'finally'          *corrected, P2, L2*
Pg 2 Line 23: 'suggested' should be 'suggest'          *corrected, P2, L18*
Pg 3 Line 17: Remove 'during recent warming' at the end of the sentence.
          *Paragraph has been removed.*
Pg 3 Line 20: Is there a citation for this? 'East AP is as vulnerable to wind dynamics as it is to temperature change'. Has a study quantified the difference in vulnerability? What vulnerability mean in this context?          *Paragraph has been removed.*
Pg 3/4 : Some citations are missing which may need including here, such as King et al 2017 and Elvidge et al 2015 which discuss föhn and melting on Larsen C.
          *Added, P3, L1-12, L29-33*
Pg 4 Line 2/3: 'These last studies taken together' doesn't read well. Perhaps change to 'Both of these studies, along with others by Elvidge et al 2015 and King et al 2017, discuss both the atmospheric. . .'. This would include the previous comment also.
          *Paragraph has been removed.*
Pg 4 Line 10: AWS is not defined yet (but is later defined on line 13/14). –
          *Corrected, P4, L5*
Pg 4 Line 14: Which satellites? Just give their names/abbreviations here.
          *Corrected, P4, L11-12*
Pg 4 Line 15 and 20: Be consistent with use of abbreviations or names. AP for example.
          *Corrected, "AP" used throughout*
Pg 4: Line 24: from what date to 2014?          *Corrected; P4, L 20*

Data and Methods:
Pg 4 Line 27/28: MAR and AWS have been defined earlier. – *Corrected, P4, L29-30*
Pg 5 Line 5: Which part of Antarctica?
          *Corrected, Reference to Terra Nova Bay, Antarctica is added P5, L5*
Pg 5, section 2.1: Where can readers can get more information about MAR, such as physics set up? Include a citation for this. What is the model top? 23 Sigma layers is very coarse (see major comments). Why was this vertical resolution used? Only 1 domain or is it nested? What is the resolution of the topography, and what dataset is used? BEDMAP2 for instance?

Corrected: Substantially more detail provided about
- Initial snow density
- Reference to previous model setup (P5, L12)
- Resolution and domain, nesting, topography
- P5, L10-37

Pg 5 Line 17: what mask? Land use? Land/sea?                    Corrected, P5 L28

Pg 5 Line 20-26: reorder this paragraph to make it clearer what each notation is. For example, line 20-23, both notations are stated, but more emphasis is put on LWC0.4. It could be split into 2 sentences, one for LWC0.4 and one for MF0.4.

Addressed in Page 6, Line 1-11(rewritten)

Pg 5 Line 25: What is the justification for this condition? The same as LWC0.4 (Tedesco et al 2007)?

The meltwater threshold is discussed in Response to Major Concerns #5

Pg 5 Line 30: change 'microwave sensors are weakly affected. . .' to 'microwave sensors are only weakly affected'.                    Corrected, P6, L26

Pg 5 line 31: after citation, change 'where' to 'whereas'. .                    Corrected, P6, L28

Pg 6, line 6/7: 'used extensively' is stated, but there is only 1 citation. Are there other important citations? The Drinkwater and Liu (2000) reference only looks at Antarctica, not Greenland.                    – additional citations added, P7, L 4-5

Pg 6, equation 1: what is Tc? The new paragraph is rewritten for clarity (P 7, L 32-36)

Pg 6: active and passive microwave: what are the spatial resolutions of the satellites? To allow some comparison with the 10km resolution of MAR.

P6, L 34; P7 L 21

Pg 7, line 2: confused what 'here' means in this context. For this location?

The new paragraph is rewritten for clarity (P 7, L 32-36)

Pg 7, line 3: 'zwa is based on the winter mean threshold'. Threshold of what? Winter mean air temperature?

The new paragraph is rewritten for clarity (P 7, L 32-36)

Pg 7, line 8: pressure observations are not mentioned here but they are in line 27 onwards.

P8 L6,7 explicitly explains that the surface pressure comparison is provided in the supplemental material

Pg 7, line 19: what is meant by 'expected'? This is also used in terms of wind speed later in the paper, and I don't understand its use.

I have added an explanation for this in text, I am using the term "expected value" (derived from the shape and scale parameters in the Weibull fit) interchangeably with the term "predicted mean" or "1ˢᵗ moment" to differentiate it from an arithmetic mean

P7 L17,18

Pg 7, line 27: What is meant by 'estimating' pressure from the AWS? Is pressure observed by the AWS or not? Pressure is also not mentioned elsewhere in the paper, so if it is not used, remove it. P8 L6,7 explicitly explains that the surface pressure comparison is provided in the supplemental material. Avoiding the word "estimated" P8, L 25

Pg 7, line 28: remove 'a' from 'also estimated at a approximately. . .' – corrected, P8, L 26

Pg 7, line 29: your lowest model level is 2m but only 23 sigma levels are used- this is very coarse. See major comments above. Are 2m diagnostics output from MAR? As you could use these instead of taking it from the lowest model level, if this should

change when you run the sensitivity study for varying the number of vertical levels.

2m values for P were not available in this model version (but will be in future model runs)

**Results:**

Pg 8, line 1: 'assess the extent to which each station is representative of larger scale climate variability'. Even though AWS14/Larsen and AWS15 are so close together? Do they have a different extent?

It is not *very* different but there's a slightly greater correlation in AWS15 to southerly regions as well as to the other side of the AP which is enough to be more affected by southwesterly flow

Pg 8, line 13: keep consistent with abbreviations. – Corrected, P9, L10

Pg 9, line 3/4: you state coordinates/latitudes in the text but there are no coordinates on your Figure 2 plots. Include coordinates on the plots.

-Corrected, now Fig. 5

Pg 9, line 19: 'data sources ad secondarily' should read 'data sources and secondarily'.

Rewritten, P11, L 29

Pg 9, line 19: What are the spatial resolutions of the data sources? You mention this, but then don't go into it any further. However, you mention the depths presumed for melt water content and then discuss it for the next paragraph. Perhaps more information on the spatial resolutions is needed.

Added explanation in text, P11, L26-29

Pg 9, line 28: Give some examples of these 'low' melt occurrence regions. From elevation information in supplement table 1, they aren't on the ice shelf, are they on the main spine of the AP?

-Addressed, P 12, L2-3

Region also coincides with NL region, described: P12, L 34-36

Pg 9, line 28: heterogeneous in what way? Elevation? Surface type? P12, L1,2

Pg 9, line 37: what is 'N column'? Corrected, P12, L11

Pg 10, line 17: what is PMWAll-coincident?

Rewritten for clarity. P 12, L 27-30

Table 1 now has abbreviations

Pg 10, line 31: 'early pulse around Dec 15th', do you mean Nov 15th? As there are small pulses of melt here, and December 15th melt looks much larger.

Clarified. P10, L 8-12

Pg 11, line 25: 'during that period'. Which period? Be more specific.

Corrected, P14, L1-2

Pg 12, line 4: remove 'station' after AWS. Corrected, P14, L16

Pg 12, line 8: remind readers of MAR-R/MAR differences here.

References to MAR-R are removed for clarity because results are unchanged, but the definition for MAR-R is included in Table 1 and kept in the figure

Pg 12, line 15/16: 'demonstrate the consistency of wind biases' and 'how wind biases vary by latitude' are slightly contradictory. Are they consistent or variable?

Rewritten for clarity, P14, L23-27

g 12, line 16: remove 'whereas', as you aren't comparing AWS and MAR, as one is for low wind and the other for high wind speeds. - Rewritten for clarity, P14, L23-27

Pg 12, line 16: 'MAR is dominated by northerly winds'. . Paragraph rewritten
Pg 12, line 27/28: Might be useful to highlight which rows of the table you mean here. When comparing all times and melt times. It isn't immediately clear that 'increased N and W flows' means compared to when there all days are included.

References to table rows added throughout

Pg 12, line 34: citation style.                                    Corrected, P15, L12
Pg 12, line 36: the abbreviation Ts is used in the table for when temperature is >0degC. However, in the text you say that when 2m-temperatures exceed 0degC. Stick to the T2m abbreviation.                                    Corrected
Pg 13, line 5: remove extra space before -3.04.                   Paragraph rewritten
Pg 13, line 8-12: include reference to figures here.             Paragraph rewritten
Pg 13, section 'observed NE flow and observed SW flow': It needs to be clearer that when MAR has different wind directions to the observations, MAR is wrong. Especially in the case where there are large differences (NE vs NW for instance). And explain what the possible reasons are for this. Is MAR not getting the synoptic scale wind direction right? Or is there not enough blocking on the west of the AP to prevent flow over the AP when there shouldn't be? This section is a good idea to see what impact the wind direction is having in MAR, but it should also be stated that if MAR is getting something like large scale flow wrong, it might be getting other processes wrong due to this.

Paragraph rewritten due to recomputation in wind fields

Pg 14, line 1-6: In this section, it might be good to remind the reader, that in case 2, MAR is getting the wind direction wrong when compared to AWS. So that the reader can put these results into context.

Paragraph rewritten due to recomputation in wind fields

Pg 14, line 7: Using Ts abbreviation but you have only talked about air temperature and used T2m previously.     Paragraph rewritten due to recomputation in wind fields
Pg 14, line 11-13: confusing sentence. What is meant by expected?

Paragraph rewritten due to recomputation in wind fields

Pg 14, line 13: I don't think figure 6 e-h are necessary. They are not discussed as much in the text, and the information is given by the 6a-d. Similarly, figure 7 could be included into figure 6 in place of 6e-6h.

Fig. 9 now shows more general wind biases in response to the recomputation of wind fields and figures have been combined

**Discussion:**
Pg 14, line 30: remove 'in the aggregate'.                        Corrected, P.16, L22
Discussion has been rewritten for clarity
Pg 15, line 4: where should be when.               Discussion has been rewritten for clarity
Pg 15, line 6/7: Any suggestions for why there are less westerly winds in MAR?
See: Response to Major Concerns #1
Pg 15, line 17-21: considering the impact of föhn winds is prominent in the abstract and introduction, this seems like a short discussion of them. See major comments.
See: Response to Major Concerns #3
Pg 15, line 19-21: wind speed may not be the biggest issue here if MAR is unable to get wind direction right.

The corrected calculation for wind direction has altered these results considerably, and we have emphasized that wind biases account for a relatively small proportion of melt occurrence captured by satellites, but not by MAR

Pg 15, line 23-25: include references to and discussion of non-hydrostatic models that have captured föhn flow- e.g Elvidge et al, 2015 (Met UM model), Turton et al 2017 (WRF model).

See: Response to Major Concerns #3

Pg 15, general: The abstract suggests that increasing the spatial resolution of MAR or the topography in the model may improve output, but this isn't discussed in your discussion. See major comment.

See: Response to Major Concerns #1

Pg 16, line 7: Figure 8 should come earlier in the text. This is a good summary figure and could be included in page 11 where interannual variability is mentioned.

This has now been moved to Fig. 2 and Sect. 3.1 (along with a comparison with RACMO in Fig. 3)

Pg 16, line 19/20: 'melt in the NL region is particularly sensitive to föhn induced melt'. You need to support this with other studies (e.g Elvidge, et al 2015, Cape et al 2015), as your study only mentions föhn jets on the SW of the ice shelf in earlier discussion.

See: Response to Major Concerns #3

Pg 16, line 22/23: is this future work? As this study doesn't talk about large-scale atmospheric drivers at all. Or you need to support this with studies which look at largescale atmospheric patterns and their related wind patterns in this region (such as Cape et al 2015).

I've eliminated this section to discuss a paper more specifically which is currently in progress.

P19, L1-5

**Figures:**
Figure 1: include in the caption that Larsen IS and AWS14 have the same MAR grid cell, which is why they are on the same marker.                     – addressed
Figure 1: Where is the topography data from?                     - addressed
Figure 1: include coordinates.                     - addressed
Figure 2: include coordinates.                     - addressed
Figure 3/4: make insert bigger, or include it in figure 1.                     - Inserted in Fig. 2
Figure 6: make a heading over a/b 'Case 1' and over c/d 'Case 2'. I don't think anything else is gained from e-h, as they are mentioned only briefly in the text. – Adressed. Figure is combined into Fig. 8, although the recomputation of wind directions means that different (more generalized) biases are discussed
Figure 6: g and h are not described in the caption.                     – Now removed
Figure 6/7: 'yellow as only shown for g,h'. Not sure what this means, as yellow markers are used in every subplot, not just g and h.                     - addressed (with a legend included)
Figure 7: could be combined with Figure 6.                     - addressed
Figure 8: if this goes earlier in the text, then the size of the insert is sufficient for the other figures which require it.                     – This is now Fig. 2
Supplementary Figure 6: lettering is not right. There is no a-c as in the figure, and g-m are not in the caption. There are only 6 subplots, so I assume a-f is correct.

                    -    corrected

**Table:**

Table 1: Ts should be T2m, unless actual surface temperature data is being used, but is not mentioned elsewhere in the paper.

- Corrected (now table 2)

Typos:
Pg 1, Line 29: satellites should be satellite          Abstract has been changed considerably
Pg 2, line 21: comma after citation                    Corrected, P2, L16
Pg 2, line 27: comma after citation                    Corrected, P2, L23
Pg 4, line 20: umlaut missing over o in föhn          Final paragraph of Intro changed
Pg 5, line 31: comma after citation                    Corrected, P6, L28
Pg 6, line 6: remove full stop after algorithm, there is one after the citation. Corrected, P7, L3

**Response to Minor Comments Referee #2:**

Page 3 l. 29 + l. 33: use same space before unit                         – corrected, P3, L23
Page 4 l. 20: change to föhn                                             Paragraph reordered
Page 5 in section 2.1: Please mention the size of the model domain
                                                        Lat/lon boundaries added in P 5 L25
Page 5 l. 2: explain abbreviation RCM
                        Term now fully introduced in Introduction P3, L14,15
Page 7 l. 6: add space after where              Paragraph reorganized for clarity
Page 7 l.35: remove space before Wilks                           - corrected, P8, L33
Page 12 l. 34: citation with 2 brackets                    - corrected, P15, L12
Page 34 l. 34: remove second brackets (assuming related to the previous comment)
Page 18 l. 8: remove slash in Royal                        - corrected, P21, L19
Page 24 l. 5: add space before Greenland                            - corrected
Figures:
Fig. 1: Please add coordinates to the axes                              – added
Fig. 2: Please add coordinates to the axes                    – added, Now Fig. 5
Fig: 3: Please have a consistent labeling of axes throughout all the figures 1-8; variable
[unit]                                                  - corrected

Fig: 4: Same as Fig. 4                                              - corrected
Fig. 7: ended with a comma ] – corrected, Figure now combined with previous figure.

[revised manuscript text omitted]

Formatted ... [13]
Formatted ... [11]
Formatted Table ... [12]
Formatted ... [14]
Formatted ... [15]
Formatted ... [16]
Formatted ... [17]
Formatted Table ... [18]
Formatted ... [19]
Formatted ... [21]
Formatted ... [20]
Formatted ... [22]
Formatted ... [23]
Formatted ... [28]
Formatted ... [29]
Formatted ... [34]
Formatted ... [35]
Formatted ... [36]
Formatted ... [37]
Formatted ... [38]
Formatted ... [39]
Formatted ... [44]
Formatted ... [45]
Formatted ... [50]
Formatted ... [52]
Formatted ... [53]
Formatted ... [54]
Formatted ... [55]
Formatted ... [56]
Formatted ... [57]

[Figure]

**Figure 9 MAR vs AWS temperatures at the Larsen Ice Shelf AWS station for DJF from 2001-2009 for melt occurrence criteria as shown bottom-right and described in text. Wind direction biases are shown for when northerly AWS flow is reported as southerly in MAR (a) AvgT2m (b) MaxT2m, when westerly AWS flow is reported as easterly in MAR (c) AvgT2m (d) MaxT2 and when AWS and MAR both report westerly flow (e) AvgT2m (f) MaxT2m.**

**Page 1: [1] Deleted**           **Rajashree Datta**           **5/7/18 7:30:00 AM**

The underestimation of föhn flow in the east of the Larsen C may potentially be resolved by removing the hydrostatic assumption in MAR or increasing spatial resolution. The underestimation of southwesterly flow in particular may be reduced by using higher-resolution topography.

**Page 3: [2] Deleted**           **Rajashree Datta**           **5/7/18 7:34:00 AM**

This is primarily caused by more exposure to open water in combination with prevailing westerly winds on the west AP and southerly winds on the east AP. Moreover, when strong westerly winds cross the bisecting mountain range of the AP (Fig. 1), the resulting föhn winds can produce pulses of warming on the East Antarctic Peninsula's ice shelves (Marshall, 2007). Föhn winds are a warm, dry air flow on the lee slopes of a mountain range (Beran 1967). This resultant warming can be produced by four main mechanisms. Elvidge (2016) uses a modeling approach to trace four physical processes that occur during föhn flow in the East AP, namely *isentropic drawdown* (sourcing of föhn air from higher altitudes), *latent heating* and *precipitation* (where cooling during uplift on the windward side promotes precipitation), *mechanical mixing* (turbulent sensible heating and drying of low-level flow) and *radiative heating* (where cloudless conditions on the lee side increase the availability of shortwave radiation for heating). The relative importance of each of these mechanisms for surface melt has been shown to be related to the source of föhn flow in the East AP (Elvidge et al., 2015; Grosvenor et al., 2014). For example, southwesterly föhn jets descending from gap flow (from lower-elevation passages in the mountain range) have been shown to be cooler and moister than surrounding föhn flow descending from higher elevations (Elvidge et al., 2015). Recent warming in the East AP has been linked to an increase in föhn winds during recent warming (Cape et al., 2015), which were possibly related to an increase in the speed of warm northwesterly winds which have been associated with positive phases of the Southern Annular Mode (SAM), (Van den Broeke and Van Lipzig, 2003). Because melt in the East AP is as vulnerable to wind dynamics as it is to regional temperature changes, an accurate depiction of föhn flow is crucial for accurate estimates of meltwater production.

**Page 3: [3] Moved to page 3 (Move #1)**           **Rajashree Datta**           **5/7/18 7:53:00 AM**

Observation-based studies on the formation of melt ponds in the Cabinet Inlet portion of the Larsen C Ice Shelf have focused on the response to föhn winds (Luckman et al., 2014) and the formation of subsurface ice (Hubbard et al., 2016). These last studies taken together discuss both the atmospheric drivers for melt as well as the effects on the ice shelf within our region of interest, but are necessarily limited to a small region where observations are available. By contrast, spaceborne satellites allow us to estimate surface melt occurrence and meltwater production over the entire AP, complementing *in-situ* data. The combination of satellite-based and *in situ* data provide an excellent toolset for model validation.

**Page 15: [4] Deleted**           **Microsoft Office User**           **6/13/18 12:12:00 AM**

Fig. 8 shows wind frequency distributions during the summer season, color-coded for wind direction as represented by the pie graph at the right. We note that AWS data are 3-hourly averages and ERA-Interim are 6-hourly averages for wind speed and direction, while MAR produced daily-averaged outputs. For this reason, a direct comparison between Weibull parameters derived from MAR vs AWS data is not fully justified. The Larsen Ice Shelf

AWS has full temporal coverage during the QuikSCAT period while AWS14 and AWS15 were installed after termination of the QuikSCAT mission. These last two stations are used in this study to demonstrate that (a) similar wind biases persisted after the QuikSCAT period at multiple locations, as AWS 14 the Larsen Ice Shelf AWSs are co-located to the same MAR grid cell and that (b) wind biases vary slightly by latitude, AWS15 being located slightly to the south.

Both MAR and AWS are dominated by northerly winds at lower windspeeds (in yellow and blue) although AWS data shows a greater frequency of southwesterly winds when windspeeds are higher (> 8 m/s). This is especially relevant at the southern AWS15, where modeled temperature correlates with a larger portion of the southern Larsen C Ice Shelf than for AWS14 (Supplemental Fig. S7). All AWSs show more southwesterly flow and slightly more northwesterly flow than either MAR-R or MAR, which show a substantially higher percentage of easterly flow instead, a trend which is more pronounced at the southernmost AWS15 (Fig. 7i,j). ERA-Interim reports substantially more northwesterly flow than either AWS or MAR and a smaller proportion of southwesterly flow in the 180°- 225° range (especially at the southernmost AWS15 location), although the proportion of easterly flow is similar to that reported by AWSs. We note that although ERA-Interim has been shown to reproduce the basic structure of föhn flow (Grosvenor et al., 2014), the resolution may be too coarse to adequately capture southwesterly gap flow here. As discussed further in Sect. 5, westerly flow towards the stations used in this study may be strongly affected by the fine-scale representation of topography (which is coarse in ERA-Interim) and the lowered orographic barrier in the northwest in ERA-Interim may contribute to the enhanced northwesterly flow shown here.

Specifically, at the Larsen Ice Shelf AWS location, both AWS and MAR reports dominant northeasterly flow (Table 2, rows 4,8, col2). However, the AWS reports slightly more flow which is either southwesterly (28.9% vs. 23.2% in MAR) or northwesterly (19.3% vs. 14.1% in MAR) while MAR reports more southeasterly flow overall (23.5% vs. 17.4% in AWS). Melt occurrence (from PMW and QS) is observed primarily when AWS-observed flow is northeasterly (0°-90°) or southwesterly (180°-270°), with QS(PMW) reporting that 36%(42%) northeasterly flow and 29%(26%) southwesterly flow. On days when MAR reports melt (Table 2, rows 19,20), southeasterly flow in MAR is even more dominant (but declines at the AWS) while northwesterly flow decreases (while it increases at the AWS). The bias towards easterly flow affects 26% of all days and 10% of melt days in MAR, 21%(18%) of all days where QS(PMW) report melt occurrence, but only 8%(9%) of days where PMW(QS) melt occurrence is not also captured by MAR. Similarly, the bias towards southerly flow captures 26% of all days and 8% of melt days in MAR, 13%(15%) of days where QS(PMW) report melt occurrence, but only 6%(6%) of days when PMW(QS) melt occurrence is not also reported by MAR. Most notably, for 4% of all melt days in MAR, AWS reports southwesterly winds while MAR reports southeasterly winds and this bias accounts for 3%(4%) of days when PMW(QS) report melt but MAR does not. In summary, despite biases in wind directions reported by MAR, the overall impact on melt occurrence is fairly limited according to comparisons with satellite estimates. Within the next section we prominent wind direction biases in greater detail.

[revised manuscript text omitted]

Left, Line spacing: single

| Page 38: [12] Formatted Table | Microsoft Office User | 5/19/18 10:57:00 AM |
|---|---|---|

Formatted Table

| Page 38: [13] Formatted | Microsoft Office User | 5/20/18 9:59:00 AM |
|---|---|---|

Line spacing: single

| Page 38: [14] Formatted | Microsoft Office User | 5/20/18 11:26:00 AM |
|---|---|---|

Font color: Text 1

| Page 38: [15] Formatted | Microsoft Office User | 5/19/18 10:55:00 AM |
|---|---|---|

Left, Line spacing: single

| Page 38: [16] Formatted | Microsoft Office User | 5/19/18 10:52:00 AM |
|---|---|---|

Line spacing: single

| Page 38: [17] Formatted | Microsoft Office User | 5/19/18 10:55:00 AM |
|---|---|---|

Left, Line spacing: single

| Page 38: [18] Formatted Table | Microsoft Office User | 5/19/18 10:57:00 AM |
|---|---|---|

Formatted Table

| Page 38: [19] Formatted | Microsoft Office User | 5/19/18 10:52:00 AM |
|---|---|---|

Line spacing: single

| Page 38: [20] Formatted | Microsoft Office User | 5/19/18 10:55:00 AM |
|---|---|---|

Left, Line spacing: single

| Page 38: [21] Formatted | Microsoft Office User | 5/19/18 10:52:00 AM |
|---|---|---|

Line spacing: single

| Page 38: [22] Formatted | Microsoft Office User | 5/19/18 10:55:00 AM |
|---|---|---|

Left, Line spacing: single

| Page 38: [23] Formatted | Microsoft Office User | 5/19/18 10:52:00 AM |
|---|---|---|

Line spacing: single

| Page 38: [24] Deleted | Microsoft Office User | 5/20/18 10:49:00 AM |
|---|---|---|

3.49

| Page 38: [24] Deleted | Microsoft Office User | 5/20/18 10:49:00 AM |
|---|---|---|

3.49

| Page 38: [25] Deleted | Microsoft Office User | 5/20/18 10:56:00 AM |
|---|---|---|

4.21

| Page 38: [25] Deleted | Microsoft Office User | 5/20/18 10:56:00 AM |
|---|---|---|
| 4.21 | | |
| Page 38: [25] Deleted | Microsoft Office User | 5/20/18 10:56:00 AM |
| 4.21 | | |
| Page 38: [26] Deleted | Microsoft Office User | 5/20/18 11:00:00 AM |
| 3. | | |
| Page 38: [26] Deleted | Microsoft Office User | 5/20/18 11:00:00 AM |
| 3. | | |
| Page 38: [26] Deleted | Microsoft Office User | 5/20/18 11:00:00 AM |
| 3. | | |
| Page 38: [27] Deleted | Microsoft Office User | 5/20/18 11:03:00 AM |
| 4. | | |
| Page 38: [27] Deleted | Microsoft Office User | 5/20/18 11:03:00 AM |
| 4. | | |
| Page 38: [27] Deleted | Microsoft Office User | 5/20/18 11:03:00 AM |
| 4. | | |
| Page 38: [27] Deleted | Microsoft Office User | 5/20/18 11:03:00 AM |
| 4. | | |
| Page 38: [28] Formatted | Microsoft Office User | 5/19/18 10:55:00 AM |
| Left, Line spacing:  single | | |
| Page 38: [29] Formatted | Microsoft Office User | 5/19/18 10:52:00 AM |
| Line spacing:  single | | |
| Page 38: [30] Deleted | Microsoft Office User | 5/20/18 10:49:00 AM |
| 3.82 | | |
| Page 38: [30] Deleted | Microsoft Office User | 5/20/18 10:49:00 AM |
| 3.82 | | |
| Page 38: [30] Deleted | Microsoft Office User | 5/20/18 10:49:00 AM |
| 3.82 | | |
| Page 38: [31] Deleted | Microsoft Office User | 5/19/18 10:46:00 AM |
| 5.10 | | |
| Page 38: [31] Deleted | Microsoft Office User | 5/19/18 10:46:00 AM |
| 5.10 | | |
| Page 38: [31] Deleted | Microsoft Office User | 5/19/18 10:46:00 AM |
| 5.10 | | |
| Page 38: [32] Deleted | Microsoft Office User | 5/20/18 11:00:00 AM |
| 4.41 | | |
| Page 38: [32] Deleted | Microsoft Office User | 5/20/18 11:00:00 AM |
| 4.41 | | |
| Page 38: [33] Deleted | Microsoft Office User | 5/19/18 10:49:00 AM |
| 4.44 | | |

| Page 38: [33] Deleted | Microsoft Office User | 5/19/18 10:49:00 AM |
|---|---|---|

4.44

| Page 38: [34] Formatted | Microsoft Office User | 5/19/18 10:55:00 AM |
|---|---|---|

Left, Line spacing:  single

| Page 38: [35] Formatted | Microsoft Office User | 5/19/18 10:52:00 AM |
|---|---|---|

Line spacing:  single

| Page 38: [36] Formatted | Microsoft Office User | 5/19/18 10:55:00 AM |
|---|---|---|

Left, Line spacing:  single

| Page 38: [37] Formatted | Microsoft Office User | 5/19/18 10:52:00 AM |
|---|---|---|

Line spacing:  single

| Page 38: [38] Formatted | Microsoft Office User | 5/19/18 10:55:00 AM |
|---|---|---|

Left, Line spacing:  single

| Page 38: [39] Formatted | Microsoft Office User | 5/19/18 10:52:00 AM |
|---|---|---|

Line spacing:  single

| Page 38: [40] Deleted | Microsoft Office User | 5/20/18 10:52:00 AM |
|---|---|---|

4.04

| Page 38: [40] Deleted | Microsoft Office User | 5/20/18 10:52:00 AM |
|---|---|---|

4.04

| Page 38: [41] Deleted | Microsoft Office User | 5/20/18 10:57:00 AM |
|---|---|---|

| Page 38: [41] Deleted | Microsoft Office User | 5/20/18 10:57:00 AM |
|---|---|---|

| Page 38: [42] Deleted | Microsoft Office User | 5/20/18 11:00:00 AM |
|---|---|---|

3.48

| Page 38: [42] Deleted | Microsoft Office User | 5/20/18 11:00:00 AM |
|---|---|---|

3.48

| Page 38: [43] Deleted | Microsoft Office User | 5/20/18 11:04:00 AM |
|---|---|---|

4.4

| Page 38: [43] Deleted | Microsoft Office User | 5/20/18 11:04:00 AM |
|---|---|---|

4.4

| Page 38: [43] Deleted | Microsoft Office User | 5/20/18 11:04:00 AM |
|---|---|---|

4.4

| Page 38: [44] Formatted | Microsoft Office User | 5/19/18 10:55:00 AM |
|---|---|---|

Left, Line spacing:  single

| Page 38: [45] Formatted | Microsoft Office User | 5/19/18 10:52:00 AM |
|---|---|---|

Line spacing:  single

| Page 38: [46] Deleted | Microsoft Office User | 5/20/18 10:54:00 AM |
|---|---|---|
| 4 | | |

| Page 38: [46] Deleted | Microsoft Office User | 5/20/18 10:54:00 AM |
|---|---|---|
| 4 | | |

| Page 38: [46] Deleted | Microsoft Office User | 5/20/18 10:54:00 AM |
|---|---|---|
| 4 | | |

| Page 38: [46] Deleted | Microsoft Office User | 5/20/18 10:54:00 AM |
|---|---|---|
| 4 | | |

| Page 38: [47] Deleted | Microsoft Office User | 5/20/18 10:57:00 AM |
|---|---|---|
| 4 | | |

| Page 38: [47] Deleted | Microsoft Office User | 5/20/18 10:57:00 AM |
|---|---|---|
| 4 | | |

| Page 38: [47] Deleted | Microsoft Office User | 5/20/18 10:57:00 AM |
|---|---|---|
| 4 | | |

| Page 38: [48] Deleted | Microsoft Office User | 5/20/18 11:01:00 AM |
|---|---|---|
| 3.90 | | |

| Page 38: [48] Deleted | Microsoft Office User | 5/20/18 11:01:00 AM |
|---|---|---|
| 3.90 | | |

| Page 38: [49] Deleted | Microsoft Office User | 5/20/18 11:04:00 AM |
|---|---|---|
| 5 | | |

| Page 38: [49] Deleted | Microsoft Office User | 5/20/18 11:04:00 AM |
|---|---|---|
| 5 | | |

| Page 38: [49] Deleted | Microsoft Office User | 5/20/18 11:04:00 AM |
|---|---|---|
| 5 | | |

| Page 38: [49] Deleted | Microsoft Office User | 5/20/18 11:04:00 AM |
|---|---|---|
| 5 | | |

| Page 38: [50] Formatted | Microsoft Office User | 5/19/18 10:55:00 AM |
|---|---|---|
| Left, Line spacing: single | | |

| Page 38: [51] Deleted | Microsoft Office User | 5/20/18 11:05:00 AM |
|---|---|---|
| **eratur** | | |

| Page 38: [51] Deleted | Microsoft Office User | 5/20/18 11:05:00 AM |
|---|---|---|
| **eratur** | | |

| Page 38: [52] Formatted | Microsoft Office User | 5/19/18 10:52:00 AM |
|---|---|---|
| Line spacing: single | | |

| Page 38: [53] Formatted | Microsoft Office User | 5/19/18 10:55:00 AM |
|---|---|---|
| Left, Line spacing: single | | |

| Page 38: [54] Formatted | Microsoft Office User | 5/19/18 10:52:00 AM |
|---|---|---|
| Line spacing: single | | |

| Page 38: [55] Formatted | Microsoft Office User | 5/19/18 10:55:00 AM |
|---|---|---|

Left, Line spacing:  single

| Page 38: [56] Formatted | Microsoft Office User | 5/19/18 10:52:00 AM |
|---|---|---|

Line spacing:  single

| Page 38: [57] Formatted | Microsoft Office User | 5/19/18 10:55:00 AM |
|---|---|---|

Left, Line spacing:  single

| Page 38: [58] Formatted | Microsoft Office User | 5/19/18 10:52:00 AM |
|---|---|---|

Line spacing:  single

| Page 38: [59] Formatted | Microsoft Office User | 5/19/18 10:55:00 AM |
|---|---|---|

Left, Line spacing:  single

| Page 38: [60] Formatted | Microsoft Office User | 5/19/18 10:52:00 AM |
|---|---|---|

Line spacing:  single

| Page 38: [61] Formatted | Microsoft Office User | 5/19/18 10:55:00 AM |
|---|---|---|

Left, Line spacing:  single

| Page 38: [62] Formatted | Microsoft Office User | 5/19/18 10:52:00 AM |
|---|---|---|

Line spacing:  single

| Page 38: [63] Formatted | Microsoft Office User | 5/19/18 10:55:00 AM |
|---|---|---|

Left, Line spacing:  single

| Page 38: [64] Formatted | Microsoft Office User | 5/19/18 10:52:00 AM |
|---|---|---|

Line spacing:  single

| Page 38: [65] Formatted | Microsoft Office User | 5/20/18 11:26:00 AM |
|---|---|---|

Font color: Text 1

| Page 38: [66] Formatted | Microsoft Office User | 5/19/18 10:55:00 AM |
|---|---|---|

Left, Line spacing:  single

| Page 38: [67] Formatted | Microsoft Office User | 5/19/18 10:52:00 AM |
|---|---|---|

Line spacing:  single

| Page 38: [68] Formatted | Microsoft Office User | 5/19/18 10:55:00 AM |
|---|---|---|

Left, Line spacing:  single

| Page 38: [69] Formatted Table | Microsoft Office User | 5/19/18 10:57:00 AM |
|---|---|---|

Formatted Table

| Page 38: [70] Formatted | Microsoft Office User | 5/19/18 10:52:00 AM |
|---|---|---|

Line spacing:  single

| Page 38: [71] Formatted | Microsoft Office User | 5/19/18 10:55:00 AM |
|---|---|---|

Left, Line spacing:  single

| **Page 38: [72] Formatted** | **Microsoft Office User** | **5/19/18 10:52:00 AM** |
| --- | --- | --- |

Line spacing:  single

| **Page 38: [73] Formatted** | **Microsoft Office User** | **5/19/18 10:55:00 AM** |
| --- | --- | --- |

Left, Line spacing:  single

| **Page 38: [74] Formatted** | **Microsoft Office User** | **5/19/18 10:52:00 AM** |
| --- | --- | --- |

Line spacing:  single

| **Page 38: [75] Formatted** | **Microsoft Office User** | **5/19/18 10:55:00 AM** |
| --- | --- | --- |

Left, Line spacing:  single

| **Page 38: [76] Formatted** | **Microsoft Office User** | **5/19/18 10:52:00 AM** |
| --- | --- | --- |

Line spacing:  single

| **Page 38: [77] Formatted** | **Microsoft Office User** | **5/19/18 10:55:00 AM** |
| --- | --- | --- |

Left, Line spacing:  single

| **Page 38: [78] Formatted** | **Microsoft Office User** | **5/19/18 10:52:00 AM** |
| --- | --- | --- |

Line spacing:  single

| **Page 38: [79] Formatted** | **Microsoft Office User** | **5/19/18 10:55:00 AM** |
| --- | --- | --- |

Left, Line spacing:  single

| **Page 38: [80] Formatted** | **Microsoft Office User** | **5/19/18 10:52:00 AM** |
| --- | --- | --- |

Line spacing:  single

| **Page 38: [81] Formatted** | **Microsoft Office User** | **5/19/18 10:55:00 AM** |
| --- | --- | --- |

Left, Line spacing:  single

| **Page 38: [82] Formatted** | **Microsoft Office User** | **5/19/18 10:52:00 AM** |
| --- | --- | --- |

Line spacing:  single

| **Page 38: [83] Formatted** | **Microsoft Office User** | **5/19/18 10:55:00 AM** |
| --- | --- | --- |

Left, Line spacing:  single

| **Page 38: [84] Formatted** | **Microsoft Office User** | **5/19/18 10:52:00 AM** |
| --- | --- | --- |

Line spacing:  single

| **Page 38: [85] Deleted** | **Microsoft Office User** | **5/19/18 12:15:00 PM** |
| --- | --- | --- |

| **Page 38: [85] Deleted** | **Microsoft Office User** | **5/19/18 12:15:00 PM** |
| --- | --- | --- |

| **Page 38: [86] Deleted** | **Microsoft Office User** | **5/19/18 12:25:00 PM** |
| --- | --- | --- |

1.31

| **Page 38: [86] Deleted** | **Microsoft Office User** | **5/19/18 12:25:00 PM** |
| --- | --- | --- |

1.31

| Page 39: [87] Deleted | Rajashree Datta | 5/21/18 6:20:00 PM |
|---|---|---|

**Missing Northerlyflow    Missing Westerly flow**

| Page 39: [88] Deleted | Microsoft Office User | 5/19/18 4:27:00 PM |
|---|---|---|

[Figure]

| Page 39: [89] Deleted | Microsoft Office User | 5/19/18 4:28:00 PM |
|---|---|---|

MAR: 4.31(±4.74) m/s   AWS: 4.81 (±7.44) m/s